Lu *et al. Genome Biology* (2021) 22:160

RESEARCH

# Genome-wide DNA mutations in Arabidopsis plants after multigenerational exposure to high temperatures

Zhaogeng Lu[1,2], Jiawen Cui[1], Li Wang[1], Nianjun Teng[3], Shoudong Zhang[4], Hon-Ming Lam[4], Yingfang Zhu[5], Siwei Xiao[6], Wensi Ke[6], Jinxing Lin[7], Chenwu Xu[2*] and Biao Jin[1*]

* Correspondence: cwxu@yzu.edu.cn; bjin@yzu.edu.cn
[2]Key Laboratory of Plant Functional Genomics of the Ministry of Education, Agricultural College of Yangzhou University, Yangzhou, China
[1]College of Horticulture and Plant Protection, Yangzhou University, Yangzhou, China
Full list of author information is available at the end of the article

## Abstract

**Background:** Elevated temperatures can cause physiological, biochemical, and molecular responses in plants that can greatly affect their growth and development. Mutations are the most fundamental force driving biological evolution. However, how long-term elevations in temperature influence the accumulation of mutations in plants remains unknown.

**Results:** Multigenerational exposure of *Arabidopsis* MA (mutation accumulation) lines and MA populations to extreme heat and moderate warming results in significantly increased mutation rates in single-nucleotide variants (SNVs) and small indels. We observe distinctive mutational spectra under extreme and moderately elevated temperatures, with significant increases in transition and transversion frequencies. Mutation occurs more frequently in intergenic regions, coding regions, and transposable elements in plants grown under elevated temperatures. At elevated temperatures, more mutations accumulate in genes associated with defense responses, DNA repair, and signaling. Notably, the distribution patterns of mutations among all progeny differ between MA populations and MA lines, suggesting that stronger selection effects occurred in populations. Methylation is observed more frequently at mutation sites, indicating its contribution to the mutation process at elevated temperatures. Mutations occurring within the same genome under elevated temperatures are significantly biased toward low gene density regions, special trinucleotides, tandem repeats, and adjacent simple repeats. Additionally, mutations found in all progeny overlap significantly with genetic variations reported in 1001 Genomes, suggesting non-uniform distribution of de novo mutations through the genome.

**Conclusion:** Collectively, our results suggest that elevated temperatures can accelerate the accumulation, and alter the molecular profiles, of DNA mutations in plants, thus providing significant insight into how environmental temperatures fuel plant evolution.

**Keywords:** *Arabidopsis thaliana*, Heat, Molecular evolution, Mutation accumulation, Mutation bias, Mutation rate, Mutation spectrum, Transposable element, Warming

## Background

Mutations are the ultimate source of genetic variations within and among all species and act as the fundamental driving force of biological evolution [1, 2]. DNA sequence mutations result from single-base substitutions, small insertions and deletions (indels), and large duplications/deletions, leading to changes in the genomic sequence composition. An accurate assessment of the rates, types, and molecular spectra of these mutations within a species is critical to advancing understanding of diverse problems in evolutionary biology, such as species divergence time [3], population genetic diversity and effective population size [4–6], intra-organism mutation rate variation [7], and adaptation to specific environments [8–11]. However, mutation rates are difficult to estimate because mutations are often subject to purifying selection and drift in natural populations [2, 12]. The prevailing mutation accumulation (MA) experiments involve establishing multiple independent lines from a single progenitor and repeatedly propagating these lines for many generations, such that the effect of selection can be greatly reduced [13]. Given the effectiveness of this experimental system, MA experiments combined with whole-genome sequencing technology have been extensively conducted to investigate spontaneous mutation rates and spectra in multiple eukaryotic species, including *Chlamydomonas reinhardtii* [14, 15], *Drosophila melanogaster* [16–18], *Caenorhabditis elegans* [19, 20], *Escherichia coli* [1, 21], *Saccharomyces cerevisiae* [22], and *Arabidopsis thaliana* [23]. These MA studies have provided important reference data for estimating overall spontaneous mutation rates and the patterns underlying the molecular evolution of various species.

In nature, however, organisms often encounter various environmental stresses, and the effect of these stresses (such as elevated temperatures) on mutations has begun to receive increasing attention. In *E. coli*, accumulated genetic changes under high-temperature stress were characterized in a long-term evolution experiment with replicate populations [8]. A recent MA study on yeast grown under high temperatures also revealed accumulated mutations associated with thermotolerance [24]. However, these studies focused on unicellular organisms; few studies have examined accumulated mutations in multicellular organisms under high temperature conditions through MA experiments.

Global warming is accelerating [25], resulting in frequent occurrences of extremely high temperatures and prolonged warming. Such changes have a profound impact on plant growth and development, productivity, and adaptation [26–28]. As sessile organisms, plants have evolved sophisticated mechanisms to cope with elevated temperatures. Over the last decade, mechanisms of heat stress response and the regulation thereof, as well as heat memory and priming in plants, have been well elucidated [29–32]. Moreover, phenological and life-history traits underlying adaptive evolution in response to climate warming have also been explored in plant populations [27, 33]. However, most studies have focused on the short-term (within a generation) responses to heat stress and phenological changes due to warming; little is known about the longer-term evolutionary genetic consequences of exposure of plants to high temperatures and mild warming over successive generations.

*A. thaliana*, as a model plant, is an ideal organism for investigating MA at the genomic level. Previous MA studies have characterized DNA sequence mutations accumulating in *A. thaliana* lineages grown in non-stress [23] and high-salinity soil stress [34]

environments. Although elevated temperatures significantly affect plant growth and development, how elevated temperatures induce MA in plants is still unknown. To address this question, we combined long-term MA experiments on lines and populations with deep whole-genome sequencing to determine the genome-wide mutation rates and profiles of *A. thaliana* grown under extreme heat, moderate warming, and control conditions for 10–22 successive generations. We found significantly increased mutation rates, along with distinct mutation spectra, mutated genes, and mutation bias, under elevated temperature conditions (moderate and extreme), providing insights into plant molecular evolution under environmental warming.

## Results

### MA experiments and whole-genome sequencing

We conducted long-term MA experiments on *A. thaliana* in both single-seed descent lineages and populations grown under Control (day 23 °C / night 18 °C), Heat (day 32 °C / night 27 °C), and Warming (day 28 °C / night 23 °C) conditions (Fig. 1a, b) (see "Methods"). The elevated temperature treatments, especially Heat (32 °C), resulted in various stress symptoms such as significantly decreased leaf size, shorter siliques (Fig. 1c, d), and shorter generation times. We sequenced 35 *A. thaliana* genomes, including 15 plants from MA lines at generation 10 (G10; five plants from each treatment) and 15 plants from MA populations [five plants each from G16 (Control, A16), G19 (Warming, C19), and G22 (Heat, B22)], spanning 10–22 successive generations, as well as their ancestor genomes (five individual plants from G0). In total, approximately 165 Gb of clean reads (30 libraries) from 30 genomes of progeny, and 25 Gb of clean reads from five libraries (see "Methods") representing the genetic background of the ancestor (Additional file 1: Table S1) were obtained. For all MA lines and populations, an average of 99.68% of sequenced reads was mapped to the *A. thaliana* reference genome, with average depths of 52.5×, 49.7×, 47.4×, 42.7×, 37.4×, and 36.3× per individual in D10 (Control), E10 (Heat), F10 (Warming), A16 (Control), B22 (Heat), and C19 (Warming), respectively. Accordingly, an average of 116 Mb (96.9%) of the reference genome was accessible for variant calling (Additional file 1: Table S1). To obtain sufficient coverage of the genetic background of the ancestor, the five G0 libraries (average coverage 37.1×) were combined. This sequencing depth/coverage and number of accessible reference sites allowed for precise detection of mutations at the whole-genome level.

### Accumulated mutations and mutation rates in MA lines and populations under elevated temperatures

We obtained a total of 211 homozygous de novo mutations from MA lines under three temperature treatments (Fig. 2a and Additional file 1: Tables S2-S3), including 39 mutations (31 single-nucleotide variants [SNVs] and 8 indels) in D10 (Control), 98 mutations (69 SNVs and 29 indels) in E10 (Heat), and 74 mutations (54 SNVs and 20 indels) in F10 (Warming). Most (85.9%) of the 57 indels in the MA lines were short (1–3 bp) deletions (dels) and insertions (ins) (Fig. 2c). Furthermore, the indels of E10 (25 dels vs. 4 ins) and F10 (17 dels vs. 3 ins) showed strong biases toward dels. In addition, we also detected 376 homozygous de novo mutations in MA populations, including 70 mutations (60 SNVs and 10 indels) in A16 (Control), 183 mutations (130

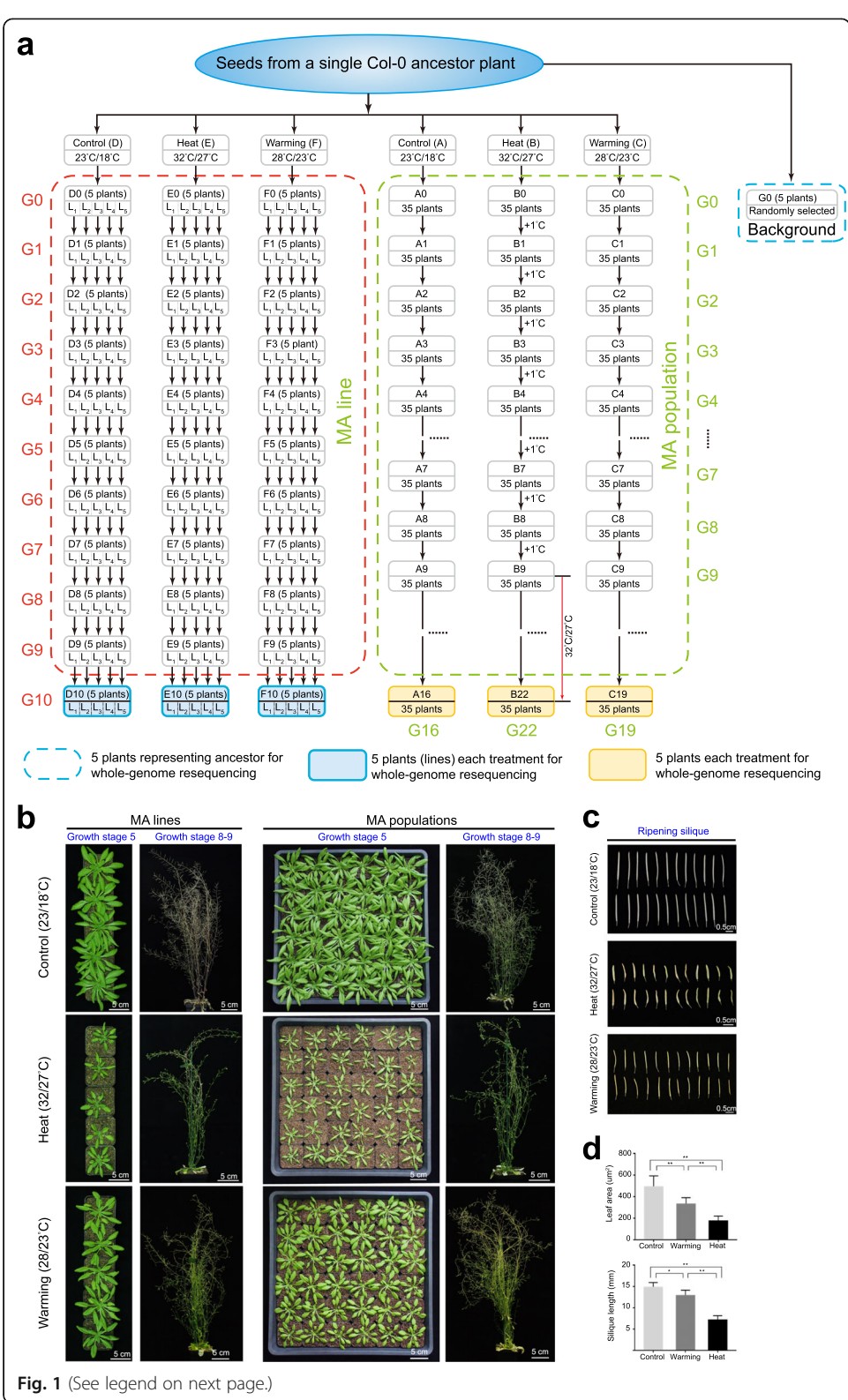

**Fig. 1** (See legend on next page.)

(See figure on previous page.)

**Fig. 1** Schematic illustration and morphological comparison of *A. thaliana* grown under Control, Heat, and Warming conditions. **a** Schematic illustration of *A. thaliana* mutation accumulation (MA) lines and populations. Two MA experiments were conducted in this study (see "Methods"). For MA line experiments, seeds from a single Col-0 ancestor plant were grown independently under Control (D), Heat (E), or Warming (F) conditions for 10 successive generations. Five 10th generation (Generation 10, G10) plants (five MA lines) from each treatment (D10, E10, F10) were used for individual whole-genome sequencing. For MA population experiments, seeds from the same ancestor plant as the MA lines were divided into three groups (~ 35 seedlings per group) and planted under Control conditions (A) for 16 generations, Warming conditions (C) for 19 generations, or Heat conditions (B) for 22 generations [the first 9 generations grown under gradual warming, i.e., increase of 1 °C per generation (from 24/18 °C to 32/27 °C [day/night]); the following 13 generations were grown at constant 32/27 °C]. Five 16th, 22th, and 19th generation plants from each treated population were also randomly selected for sequencing. To maximize coverage and provide progenitor background genetic information (reference genome sequence) for MA experiments, five individuals (G0) were combined for sequencing. Genome-sequenced plants from MA lines and populations are highlighted in yellow- and grey-shaded (blue outline) boxes, respectively; see also Additional file 1: Table S1. **b** Growth status of MA plants exposed to Control, Heat, and Warming conditions at stage 5 (bolting) and stages 8–9 (silique ripening and senescence). Leaves at stage 5 (major axis ≤ 1 cm) were sampled for DNA extraction and sequencing. Scale bar, 5 cm. **c** Ripened siliques from the Control, Heat, and Warming treatments. Scale bar, 0.5 cm. **d** Phenotypic statistics of leaf area and silique length under different temperature treatments. Leaves at stage 5 (bolting) and siliques at stage 9 were measured. The experiments were repeated three times and the data are presented as means ± standard errors of the mean (SEMs; $n = 30$). Significant differences were revealed using analysis of variance (ANOVA) with post hoc tests (*$p < 0.05$, **$p < 0.01$ vs. Control or Warming)

SNVs and 53 indels) in B22 (Heat), and 123 mutations (88 SNVs and 35 indels) in C19 (Warming) (Fig. 2b and Additional file 1: Tables S2-S3). Similar to the indels identified from MA lines, most indels in the MA populations were short (1–3 bp) and biased toward dels in B22 (42 dels vs. 11 ins) and C19 (22 dels vs. 13 ins) (Fig. 2d). Moreover, we found no novel transposable element (TE) insertion event in any MA line or population.

We further estimated the accuracy of the mutation calling pipelines using two simulation tests [14, 35]. For the first test, we simulated 600 random SNVs using six copies of reference genomes (see "Methods"). After read mapping and SNV filtering against the mutated reference genomes, our pipeline recovered 588 (98%) of 600 expected SNVs (Additional file 1: Table S4). For the second simulation test, we introduced homozygous SNVs and performed heterozygous SNV filtering, resulting in the recovery of 71–91% homozygous SNVs (Additional file 1: Table S5). To confirm our mutation calls, we experimentally examined all SNVs and indels from MA lines by Sanger sequencing. In total, 205 of 211 mutations were confirmed (six mutations were identified as PCR failures) (Additional file 1: Table S6).

We estimated the SNV mutation rate ($\mu_{SNV}$) and indel mutation rate ($\mu_{indel}$) per site per generation in the MA lines. Mutigenerational growth of *A. thaliana* under heat conditions caused significant increases relative to Control D in the average rates of SNVs [$\mu_{E\text{-}SNV} = 1.18$ (± 0.09) × $10^{-8}$ vs. $\mu_{D\text{-}SNV} = 5.28$ (± 0.95) × $10^{-9}$; two-sample $t$ test, $p = 1.0 \times 10^{-3}$] and indels [$\mu_{E\text{-}indel} = 4.94$ (± 0.50) × $10^{-9}$ vs. $\mu_{D\text{-}indel} = 1.36$ (± 0.43) × $10^{-9}$, $p = 6.3 \times 10^{-4}$; Fig. 3a]. Similarly, Warming also increased the SNV [$\mu_{F\text{-}SNV} = 9.21$ (± 0.68) × $10^{-9}$, $p = 9.9 \times 10^{-3}$] and indel [$\mu_{F\text{-}indel} = 3.41$ (± 0.71) × $10^{-9}$, $p = 0.04$] mutation rates. Furthermore, the mutation rates of dels were more than 5-fold higher than those of ins in both Heat E [$\mu_{E\text{-}del} = 4.26$ (± 0.47) × $10^{-9}$ vs. $\mu_{E\text{-}ins} = 6.82$ (± 1.70) × $10^{-10}$] and Warming F [$\mu_{F\text{-}del} = 2.90$ (± 0.88) × $10^{-9}$ vs. $\mu_{F\text{-}ins} = 5.12$ (± 3.41) × $10^{-10}$], in contrast to the lack of difference observed in Control D. The overall MA mutation

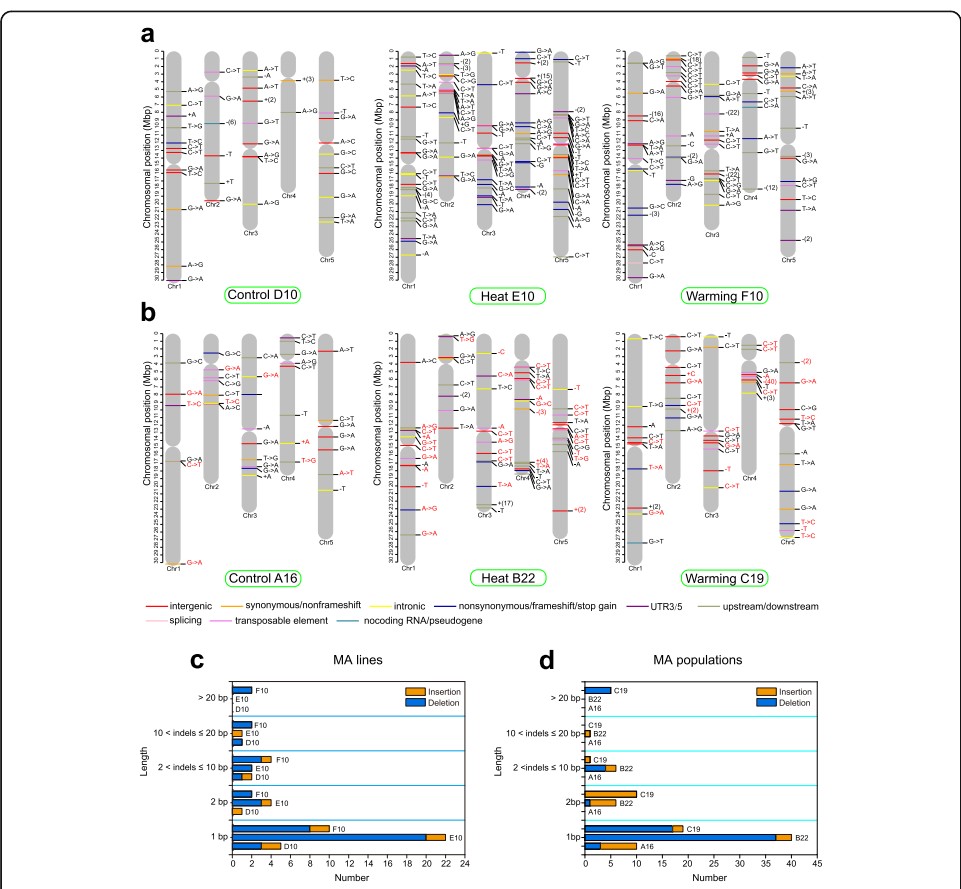

**Fig. 2** Distribution across chromosomes of de novo mutations [single-nucleotide variants (SNVs) and small insertions and deletions (indels)] detected in genomes of *Arabidopsis* from the Heat, Warming, and Control MA lines and populations. **a**,**b** Labels indicate the type of mutation; colors indicate the functional class or predicted consequence. Single-base insertions (ins) and deletions (dels) are indicated by base letters preceded by a plus and minus sign, respectively. Large ins and dels are indicated by a plus (with the number of inserted base pairs) and minus sign (with the number of deleted base pairs), respectively. Individual colors indicate intergenic region (red), intron (yellow), synonymous/non-frameshift (orange), nonsynonymous/frameshift/stop gain (blue), UTR3/5 (purple), upstream/downstream (green), splicing (pink), transposable element (violet), and noncoding/pseudogene (lake blue) mutations. Red labels in each MA population indicate the same mutations detected in at least two sequenced samples. **c**, **d** Frequencies and categories of ins and dels were determined based on their indel lengths (see also Additional file 1: Table S3)

rates (SNVs and indels) of the Heat and Warming lines were 1.67 ($\pm$ 0.06) $\times$ $10^{-8}$ ($\mu_{\text{E-total}}$) and 1.26 ($\pm$ 0.13) $\times$ $10^{-8}$ ($\mu_{\text{F-total}}$) per site per generation, approximately 2.5-fold ($p$ = 8.6 $\times$ $10^{-6}$) and 1.9-fold ($p$ = 4 $\times$ $10^{-3}$) higher than the Control [$\mu_{\text{D-total}}$ = 6.65 ($\pm$ 0.83) $\times$ $10^{-9}$], respectively (Fig. 3a). In addition, we observed significantly higher rates of total mutations and SNVs in Heat E compared to Warming F ($p$ < 0.05), whereas the difference in indel rates was not significant ($p$ = 0.1).

In parallel, we also estimated the average mutation rates in MA populations (Fig. 3b). The plants grown under Heat conditions had significantly higher SNV rates than Control plants, such as $\mu_{\text{B-SNV}}$ (Heat) = 1.03 ($\pm$ 0.06) $\times$ $10^{-8}$ vs. $\mu_{\text{A-SNV}}$ (Control) = 6.53 ($\pm$ 0.76) $\times$ $10^{-9}$ ($p$ = 1.6 $\times$ $10^{-4}$), whereas no significant difference was observed between Warming ($\mu_{\text{C-SNV}}$ = 8.08 ($\pm$ 0.63) $\times$ $10^{-9}$) and Control conditions ($p$ = 0.2). However, the total mutation rates of Heat B and Warming C were 1.45 ($\pm$ 0.09) $\times$ $10^{-8}$ ($\mu_{\text{B-total}}$)

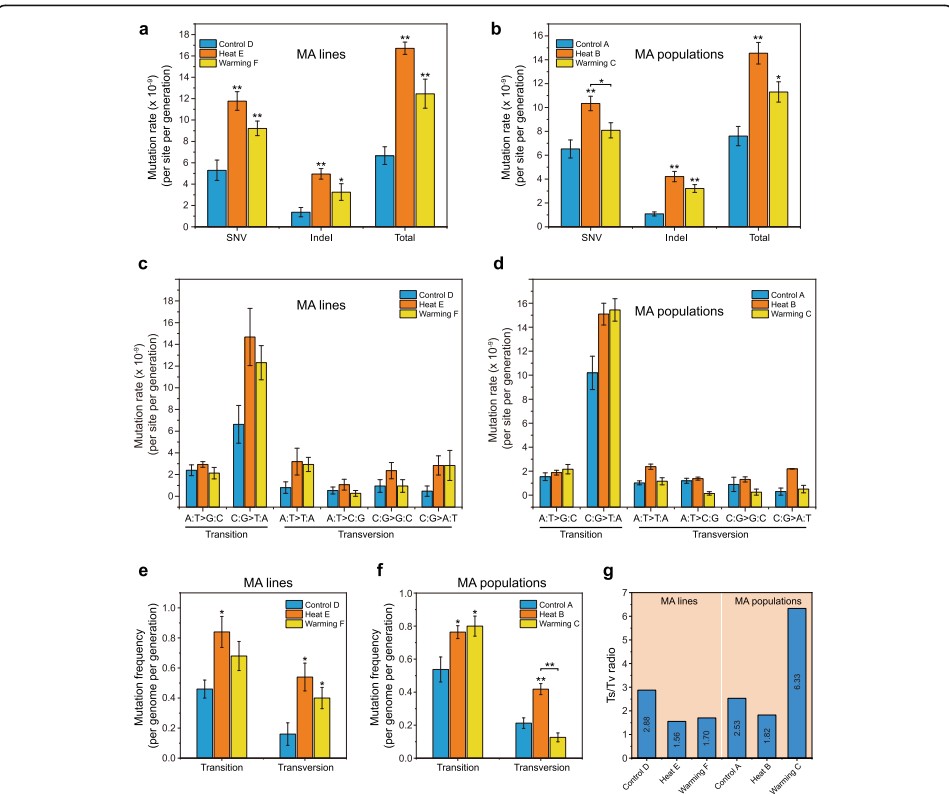

**Fig. 3** Estimation of mutation rates of observed mutations (SNVs, indels) and molecular spectra in Control, Warming, and Heat MA lines and populations. **a**, **b** SNV, indel, and total mutation rates (per site per generation) of de novo mutations in MA lines and populations subjected to different temperature treatments. Significant differences were revealed using a two-tailed Student's *t* test (*$p < 0.05$, **$p < 0.01$ compared to the Control or Warming treatments). **c**, **d** Mutation rates of different mutation types in MA lines and populations subjected to different temperature treatments. Conditional rates of each mutation type per site per generation were estimated by dividing the number of observed mutations by the number of analyzed sites capable of producing a given mutation and the number of generations of MA in each Control, Warming, and Heat lineage and population lineage. Error bars indicate SEM. **e**, **f** Mutation frequencies (per genome per generation) of transition and transversion mutations accumulated in MA lines and populations subjected to different temperature treatments. Significant differences were revealed using a two-tailed Student's *t* test (*$p < 0.05$, **$p < 0.01$ compared to the Control or Warming treatments). **g** Transition/transversion ratios (Ts/Tv) of SNVs accumulated in MA lines and populations subjected to different temperature treatments

and 1.13 (± 0.09) × $10^{-9}$ ($\mu_{\text{C-total}}$), nearly 2.0- and 1.5-fold ($p < 0.05$) higher than in Control A [$\mu_{\text{A-total}} = 7.61$ (± 0.08) × $10^{-9}$], respectively (Fig. 3b). Additionally, compared to Warming C, Heat B had significantly higher total mutation and SNV rates ($p <$ 0.05). However, the SNV rates and total mutation rates were lower in both the Heat and Warming populations than in the MA lines under elevated temperatures.

### Molecular spectra of mutations in *A. thaliana* under elevated temperatures

Base substitution mutation spectra varied after multigenerational growth of *A. thaliana* under Heat, Warming, and Control conditions. We found a strong C:G → T:A bias (driven by C → T and G → A) in six mutational spectra that commonly occurred in MA lines under all three temperature treatments (Fig. 3c); however, C:G → T:A mutations under elevated temperatures (Heat and Warming) had much higher rates

compared to Control. Furthermore, compared to Heat E ($\mu_{\text{E-C:G} \to \text{T:A}} = 1.47 \times 10^{-8}$ per site per generation), Warming F exhibited a lower C:G → T:A mutation rate ($\mu_{\text{F-C:G} \to \text{T:A}} = 1.23 \times 10^{-8}$). In addition, in Heat E and Warming F, the second most frequent substitution was A:T → T:A (mutation rate, $\mu_{\text{E-A:T} \to \text{T:A}} = 3.20 \times 10^{-9}$); however, this differed from Control D, in which the second most frequent substitution was A:T → G:C ($\mu_{\text{E-A:T} \to \text{G:C}} = 2.39 \times 10^{-9}$). In general, the mean rate of mutations occurring at C:G sites was nearly 3-fold higher than at A:T sites in Heat E and Warming F (Fig. 3c), in contrast to ~2-fold in Control D. In MA populations, we observed similar results under Heat and Warming (Fig. 3d); for example, the most frequent substitutions in Heat B and Warming C were also biased toward C:G → T:A ($\mu_{\text{B-C:G} \to \text{T:A}} = 1.50 \times 10^{-8}$; $\mu_{\text{C-C:G} \to \text{T:A}} = 1.54 \times 10^{-8}$) and were higher than in Control A ($\mu_{\text{A-C:G} \to \text{T:A}} = 1.01 \times 10^{-8}$). The second most frequent substitutions (mutation rate) in Heat B occurred at A:T → T:A ($\mu_{\text{B-A:T} \to \text{T:A}} = 2.37 \times 10^{-9}$) sites, similar to Heat MA lines. By contrast, the second most frequent substitutions in Warming C were A:T → G:C ($\mu_{\text{C-A:T} \to \text{G:C}} = 2.16 \times 10^{-9}$), somewhat different from Warming MA lines.

We calculated the transition and transversion frequencies (per genome per generation) for the three treatments. In MA lines, the transition (Ts) and transversion (Tv) frequencies in Heat E (0.84 and 0.54, respectively) and Warming F (0.68 and 0.40, respectively) showed obvious increases compared to Control D (0.46 and 0.16, respectively; Fig. 3e,f), resulting in significantly decreased Ts/Tv ratios at both elevated temperatures (Fig. 3g and Additional file 1: Table S7). Moreover, compared to Heat E, Warming F had a higher Ts/Tv ratio, which can be attributed to its lower frequencies of Ts and Tv. The Ts/Tv ratios were higher in Heat and Warming MA populations than in the MA lines (Fig. 3g). Within MA populations, the Ts/Tv ratios were significantly decreased in Heat B (1.83) compared to Control A (2.53; Fig. 3g). Nevertheless, the Warming population showed a high Ts/Tv ratio due to its higher transition and lower transversion rates relative to the Heat and Control populations.

### Mutation frequency distribution across different genomic regions in *A. thaliana* under elevated temperatures

We annotated the mutations and estimated their frequencies across different genomic regions in MA lines and populations. All MA lines showed higher mutation frequencies in intergenic regions than in genic regions. Heat E and Warming F showed > 50% increases in mutation frequencies in both genic (1.4–2.2-fold increases) and intergenic (0.5–1-fold increases) regions compared to Control D ($p < 0.05$; Fig. 4a; Additional file 1: Table S8A). Notably, within genic regions of Heat E and Warming F, higher mutation frequencies occurred in coding regions than in noncoding regions, different from Control D. The predominance of variants in coding regions of Heat E and Warming F was attributed to the disproportionate occurrence of nonsynonymous mutations (Additional file 1: Table S8A). For example, the nonsynonymous mutations in Heat E (0.26) were significantly more frequent than in Control D (0.02; $p < 0.05$). In addition, Heat E showed higher mutation frequencies in intergenic and genic regions than Warming F, but this difference was not significant ($p > 0.05$). In noncoding regions, the frequencies of intronic and untranslated region (UTR) mutations were highest in Heat E. Interestingly, more mutations occurred in transposable elements (TEs) under the Heat

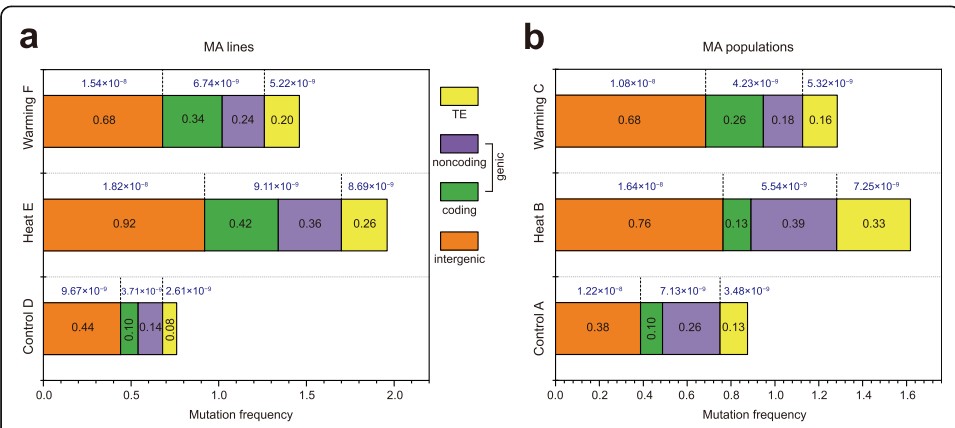

**Fig. 4** Comparison of mutation frequencies in various genomic regions among the Control, Heat, and Warming lines (**a**) and populations (**b**). The numerical values in the stacked bar chart indicate the frequencies of total mutations (SNVs and indels) in the genomic regions. The mutation frequency of each region in each sample was calculated using the formula $m = n/g$, where $n$ is the number of identified mutations and $g$ is the number of generations. Accordingly, the mean mutation frequency of each treatment (five samples) was estimated by the $\sum m/5$. Numerical values above the bars indicate SNV rates in the genomic regions. The SNV rates of each genomic region (per site per generation) were estimated by dividing the number of observed mutations by the number of analyzed sites capable of producing a given mutation and the number of generations of MA

treatment, with a significantly increased frequency in Heat E compared to Control D ($p = 0.02$). Finally, we calculated SNV rates within intergenic, genic, and TE regions (Fig. 4a), all of which increased with temperature.

In MA populations, we found that the mutation frequencies of intergenic regions and TEs in Heat B were significantly higher than those in Control A ($p < 0.01$; Fig. 4b). By contrast, the frequency of nonsynonymous mutations in Heat B (0.07) was lower than that in Warming C (0.18) (Additional file 1: Table S8B). Consistently, the mutation frequency of coding regions in Heat B (0.13) was lower than those in Warming C (0.26) (Fig. 4b). Notably, the mutation frequencies of coding regions in Heat and Warming populations were also lower than in the Heat and Warming lines, with a significantly lower frequency of nonsynonymous mutations observed in Heat B population (0.07) than in the Heat E lines (0.26) (Additional file 1: Table S8B); this indicates the stronger selection effects for nonsynonymous mutations in MA populations at high temperatures. To further investigate the selection effects on MA populations, we used the KaKs calculator to determine the ratio of nonsynonymous to synonymous substitutions (Ka/Ks ratio). The Heat E lines had a Ka/Ks ratio of 0.92, whereas the Heat B population had a Ka/Ks ratio > 1 (1.51), suggesting that the Heat MA population had been subjected to positive selection.

Nonsynonymous SNVs, gains and losses of stop codons, and indels within coding regions are likely to affect fitness [36]. Therefore, we estimated the rates of diploid genomic mutations affecting fitness under different treatments. These rates were 0.48 (± 0.1) and 0.36 (± 0.2) per generation in Heat E and Warming F, respectively, and these values were higher than those in Control D (0.13 ± 0.1; Heat *vs.* Control, $p = 0.005$; Warming *vs.* Control, $p = 0.16$). Similarly, genomic mutation rates affecting fitness were significantly higher in Heat B (0.16) and Warming C (0.24) than in Control A (0.05; $p < 0.003$). In addition, genomic mutation rates affecting fitness were lower in MA populations than MA lines.

### Mutations in functional genes of *A. thaliana* under elevated temperatures

To investigate the accumulated mutations in functional genes that may be involved in various biological processes underlying high-temperature responses, we performed Gene Ontology (GO) functional analysis of 29, 46, and 55 genes with mutations from the Control, Warming, and Heat MA lines and populations, respectively (Additional file 1: Table S3). We found that these genes were enriched in multiple related terms, including the cellular process, metabolite process, cell part, membrane, binding, and catalytic activity (Fig. 5a). In contrast, elevated temperatures resulted in the enrichment of more genes associated with the "response to stimulus," "reproductive process," "development process," and "biological regulation" terms. Kyoto Encyclopedia of Genes and Genomes (KEGG) functional analysis showed enrichment of common pathways, including "signaling transduction," "development," and "replication and repair" at elevated temperatures (Additional file 2: Fig. S1).

To further determine whether these mutations occurred at genes involved in the transcriptional response to heat and warming, we used a previously obtained RNA-seq dataset to identify potential temperature-responsive (significantly differentially expressed) transcripts among the Heat, Warming, and Control treatments (see "Methods"). Interestingly, 9 (16%) of 55 genes from Heat MA samples showed significantly differential expression between the Control and Heat treatments, and 10 (22%) of 46 genes from Warming MA samples were differentially expressed between Control and Warming (Fig. 5b). In particular, mutations occurred in two genes encoding heat-shock protein 70-17 (*HSP70-17*) and heat stress transcription factor A-1a (*HSFA1A*), which were upregulated under Heat treatment and were identified in Heat E and B, respectively.

We further focused on genes with nonsynonymous, frameshift, stop-gain SNVs, or indels (Fig. 5c). In Heat lines, in addition to *HSP70-17*, described above, a nonsynonymous mutation was found in a gene encoding fumarate hydratase 2 (*FUM2*), which is associated with respiratory metabolism. Interestingly, a mutation occurred in the gene encoding a proliferating cell nuclear antigen (PCNA) domain-containing protein (*AT4G17760*), which is associated with DNA repair. Moreover, a frameshift del and a nonsynonymous SNV in the defense-related (i.e., disease resistance) protein Toll interleukin 1 receptor-like nucleotide-binding leucine-rich repeat (TIR-NB-LRR; *AT5G48770* and *AT4G10780*) were identified in the Heat E and warming F lines, respectively. In contrast to the accumulated mutations in MA lines, we found many exonic mutations distributed in genes associated with development and signal transduction, such as those encoding the calcium-dependent lipid-binding family protein (*AT1G48090*) and WD40 repeat-like superfamily protein (*AT3G54190*), in MA populations (Fig. 5c). Notably, the mutation distribution patterns among all individuals differed significantly between MA populations and MA lines. For example, some mutations within a MA population were shared by different individuals, whereas MA line mutations were scattered widely among individuals (Fig. 5c, Additional file 1: Table S3); these results demonstrated the distinct mutation landscapes of MA populations and MA lines. Given the experimental design applied to MA populations, we speculate that these common exonic mutations probably originated from a parental individual in the same generation (not the ancestor plant), suggesting that some genetic variants are more likely to spread in populations under selective pressure over multiple generations.

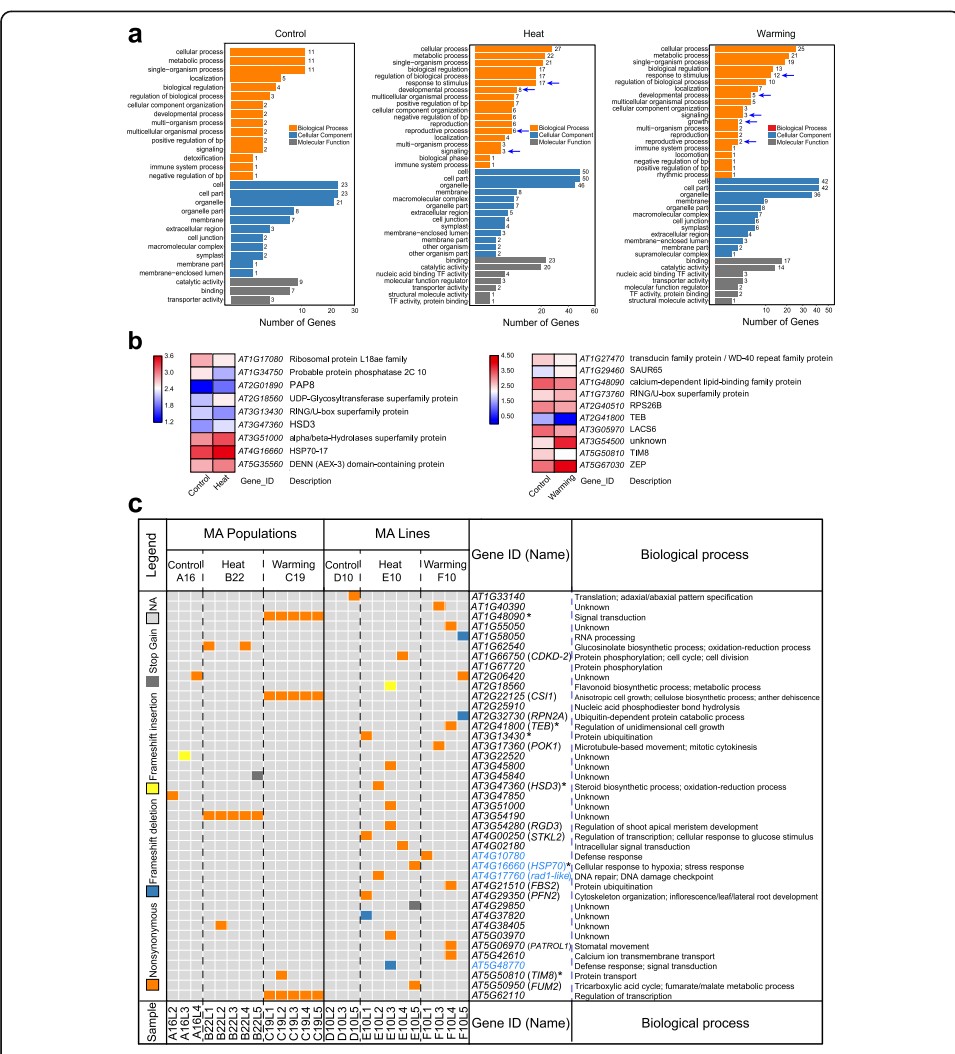

**Fig. 5** Functional enrichment of mutated genes in MA lines and populations. **a** GO enrichment of mutated genes in the Control (A and D), Heat (B and E), and Warming (C and F) treatments. The arrow indicates an important biological process. **b** Expression levels of mutated genes under Heat and Warming differed significantly from the Control (expression dataset obtained from NCBI GSE118298). Log10-transformed FPKM expression values for each treatment were visualized using a heatmap. Red color indicates a high expression level, and blue color indicates a low expression level. **c** Nonsynonymous (orange), frameshift deletion (blue), frameshift insertion (yellow), and stop gain (grey) mutations in gene coding regions of MA lines and populations. Each gene involved in a putative biological process is shown. Defense response- and DNA repair-associated genes are marked in blue, and asterisks indicate the differentially expressed genes shown in (**b**)

## Interaction between methylation and TE annotation

We conducted whole-genome bisulfite sequencing of MA lines and identified more methylated cytosines (mCs) in Heat E (10.54%) and Warming F (10.44%) than in Control D (9.78%; Additional file 1: Table S9). mCs in CG, CHG, and CHH (where H refers to A, T, or G) contexts are summarized in Supplemental Table 8. Spontaneous deamination of methylated cytosine (mC) to thymine is known to be a major source of mutations, resulting in elevated mutation rates at methylated sites [37]. We thus focused on mutations in the Control, Heat, and Warming MA lines in three contexts methylated and nonmethylated contexts. In the Heat treatment, the proportions of methylation at

mutated bases were much greater than the genome-wide occurrence of methylation in the CG (Fisher's exact test, $p = 4.58 \times 10^{-8}$), CHG (Fisher's exact test, $p = 1.92 \times 10^{-21}$), and CHH (Fisher's exact test, $p = 1.63 \times 10^{-3}$) contexts (Fig. 6b). High frequencies of methylation at mutation sites were also found in the Warming (Fisher's exact test: CG, $p = 3.36 \times 10^{-4}$; CHG, $p = 1.92 \times 10^{-15}$; CHH, $p = 0.02$) and Control (Fisher's exact test: CG, $p = 3.56 \times 10^{-12}$; CHG, $p = 2.27 \times 10^{-21}$; CHH, $p = 0.08$) treatments (Fig. 6a, c). Because methylation and TEs correlate significantly [35, 38], we further tested the main effects of methylation and TE position (two-way analysis) on mutation rates under elevated temperatures using a logistic regression model. The methylated sites and TE regions were associated positively with mutations in the Control, Heat, and Warming MA lines (Additional file 1: Table S10). In general, methylated sites within and outside TEs had higher mutation rates than did nonmethylated sites in MA lines (Additional file 2: Fig. S2). Compared with those in Control lines, methylated and nonmethylated sites in the Heat and Warming lines showed higher mutation rates regardless of location (within or outside TEs); Heat E had the highest mutation rate on methylated sites outside TEs (Fig. 6d). In addition, we observed that nonmethylated sites within TEs had a higher rate in Warming F than in Heat E, but this difference was not significant.

### Mutational bias and context effects of *A. thaliana* under elevated temperatures

We estimated the relative contributions of various genomic properties to mutation frequency, including gene density and GC content. In both MA lines and populations, we found that fewer mutations in high versus low gene density regions in all three treatments (Fig. 7a). For MA lines, the mutation rate was significantly biased toward low versus high gene density region in Heat E (*t* test, $p = 0.02$; Fig. 7b) and Warming F ($p = 0.03$). By contrast, no significant difference in mutation rate of Control D was observed between the high and low gene density regions ($p = 0.45$). This result suggests that multigenerational exposure of *A. thaliana* to high temperatures accelerates the accumulation of DNA mutations toward low gene density region compared to plants under ambient (Control) temperature. We also estimated whether GC content (per 1-kb window) affected local mutation rates in the MA lines and populations of each treatment. For all MA lines and populations, the GC contents and observed mutation rates were not well correlated (Additional file 2: Fig. S3), suggesting that GC content did not affect the mutation rate in our MA experiments.

We evaluated the effect of local sequence context on the mutation rates of A/T and G/C positions flanked by different nucleotides at either site, regardless of DNA strand orientation (for example, AAG and its complement CTT both contribute to the mutation rate in the central AT position under the category AAG). As expected, GC bases had significantly higher mutation rates than did AT bases in all treatments (*t* test, $p < 0.03$) except Control D ($p > 0.14$; Fig. 7c–e). In general, mutation rates of AT bases in all contexts were uniform for each MA experiment (*G* test, $p > 0.15$), but mutation rates of GC bases were not ($p < 0.01$), except in the Warming treatments (Warming F, $p = 0.98$; Warming C, $p = 0.44$). Moreover, the nucleotides located one position upstream or downstream had significant effects on the mutation rate in the Heat B population (*t* test, $p < 3.32 \times 10^{-4}$; Fig. 7d), whereas those in other MA lines and

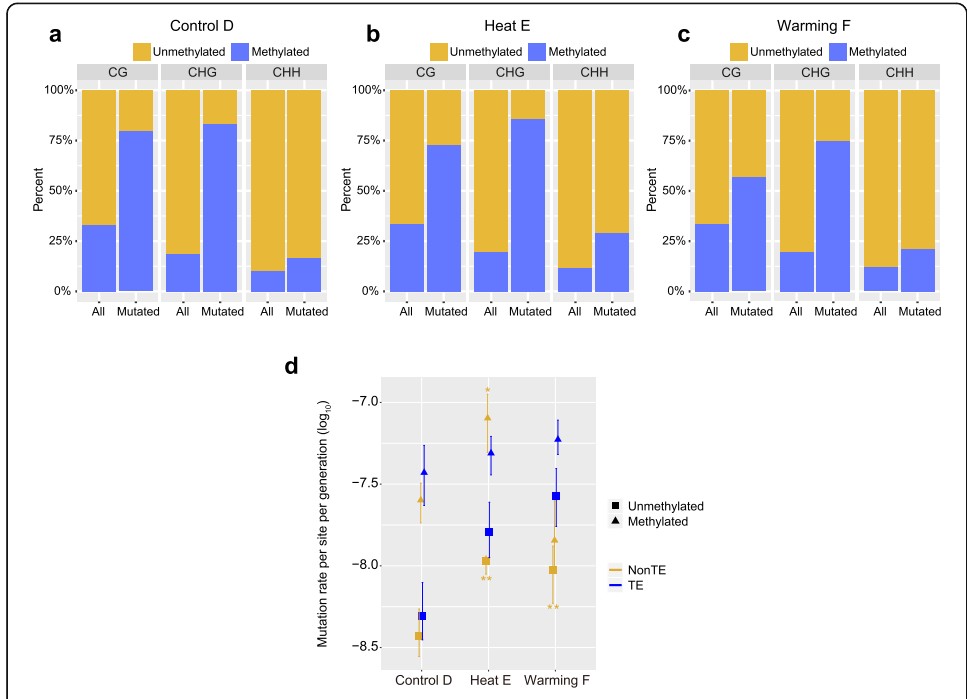

**Fig. 6** Estimation of the effects of cytosine methylation and TE region on mutation rates in the Control, Heat, and Warming MA lines. **a–c** Comparison of cytosine methylation percentages at all bases in the genome and mutated bases in the Control D (A), Heat E (B), and Warming F (C) lines. H refers to A, T, or G. The methylation percentage is much higher at mutated bases than the corresponding genome-wide occurrence for all three contexts: CG (Fisher's exact test, $p = 4.58 \times 10^{-8}$), CHG (Fisher's exact test, $p = 1.92 \times 10^{-21}$), and CHH (Fisher's exact test, $p = 1.63 \times 10^{-3}$). **d** Effects of cytosine methylation and TE region on mutation rates in the Control D (D), Heat E (E), and Warming F (F) lines. The *x* axis shows log-transformed (log10) mutation rates per site per generation. Mutation rates for non-TE and TE positions are marked in orange and blue, respectively. Mutation rates for nonmethylated and methylated CG positions are indicated with triangles and squares, respectively. Differences in mutation rates among Control D, Heat E, and Warming F lines were assessed using Student's *t* test. Error bars indicate SEMs. Asterisks indicate significant differences from Control D at $p < 0.05$ (*) and $p < 0.01$ (**), respectively

populations did not ($p > 0.05$). Of all 16 possible combinations of flanking nucleotides, GCG in Control A (two-tailed $Z$ test, $p = 5.56 \times 10^{-13}$), GCG in Heat B ($p = 7.40 \times 10^{-14}$), and CCG in Warming C ($p = 3.97 \times 10^{-6}$) had significantly higher mutation rates than did other GC contexts. In contrast to MA populations, the Control D, Heat E, and Warming F lines showed significantly higher mutation rates in the CCC ($p = 3.03 \times 10^{-4}$), CCG ($p = 0.02$), and GCT ($p = 6.24 \times 10^{-5}$) contexts than in other GC contexts (Fig. 7g, h). However, the trinucleotides CCG (or GGC) and GCG (or CGC) appeared to have high mutation rates in all MA groups, regardless of temperature treatment. In addition, we observed that almost all indels within the Heat (E10 and B22), Warming (F10 and C19), and Control groups (D10 and A16) either occurred near simple repeats or involved tandem-repeat dels and ins (Additional file 1: Table S11), suggesting that the occurrence of indels is strongly biased toward repeat sequences in *A. thaliana*.

## Comparison of de novo mutations with natural genetic variations

The 1001 Genomes Consortium (2016) reported 10,707,430 single-nucleotide polymorphisms (SNPs) and 1,424,879 indels (≤ 40 bp) in 1135 natural accessions of *A. thaliana*.

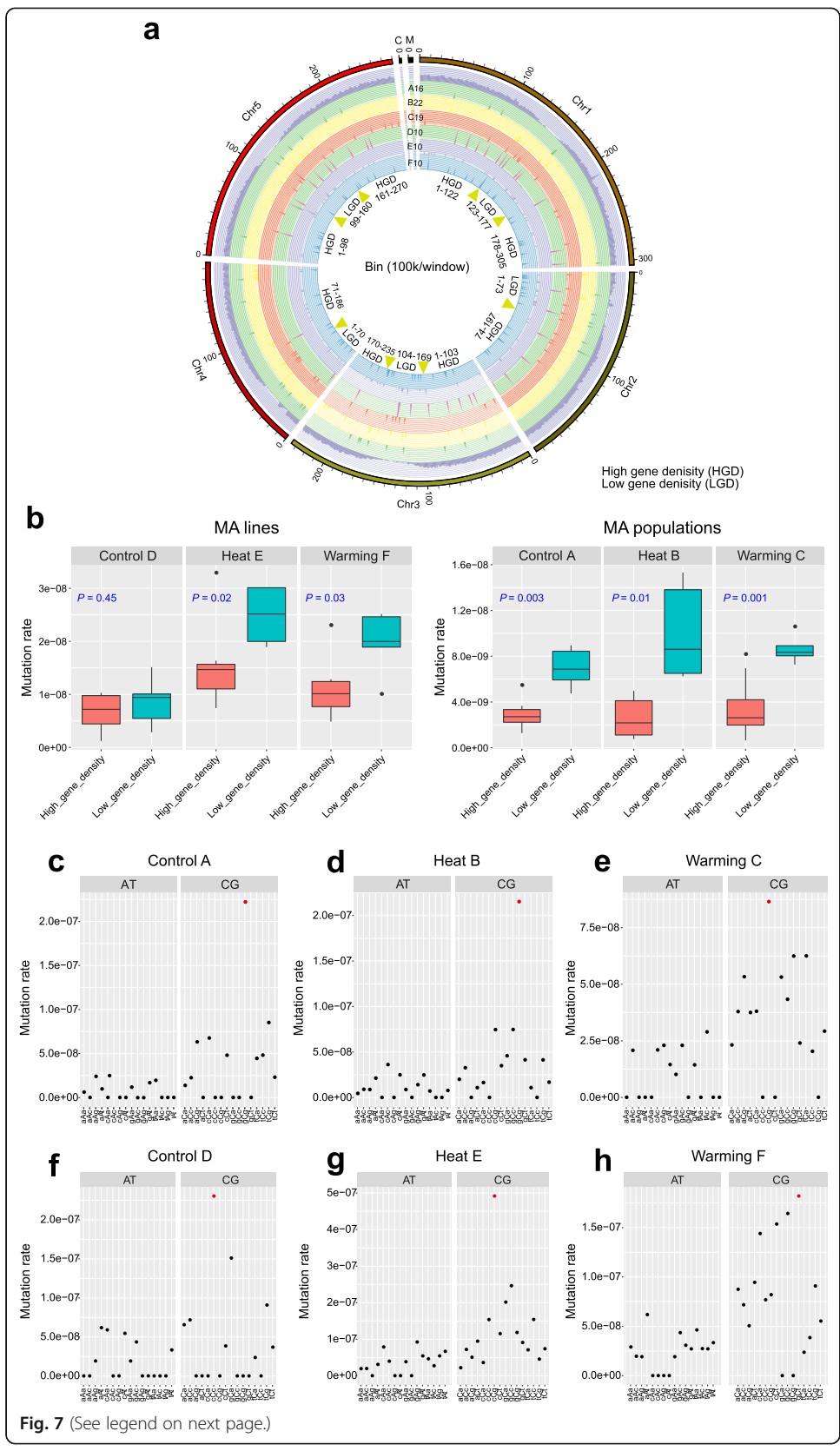

**Fig. 7** (See legend on next page.)

(See figure on previous page.)
**Fig. 7** Mutational biases of Control, Heat, and Warming lines and populations. Analysis of correlations between gene density and mutation rates across chromosomes in Control, Heat, and Warming lines and populations. **a** Distribution of mutations across *A. thaliana* chromosomes shown in a Circos plot. From outer circle to inner circle, the plot shows the chromosomes, the genes (purple bars), and mutations in A16 (green bars), B22 (yellow bars), C19 (red bars), D10 (pink bars), E10 (purple bars), and F10 (blue bars). Each chromosome is divided into multiple bins (bin size = 100 kb), which are grouped into high and low gene density regions. **b** Comparison of mutation rates between regions with high gene density and those with low gene density. Significant differences were revealed using two-tailed Student's *t* test ($p < 0.05$, high vs. low gene density regions). **c–h** Neighbor-dependent mutation rates at AT and GC bases estimated for the Control, Heat, and Warming MA lines (**c–e**) and populations (**f–h**). The trinucleotide context-dependent mutation rate is shown for each treatment. The *x* axis shows the focal nucleotides (uppercase, mutation site) and immediate flanking nucleotides (lowercase), regardless of strand orientation (e.g., the tAt class includes the overall mutation rate at tAt and aTa sites). For each treatment, the mutation rates of G/C bases were generally elevated relative to those of A/T bases. Red dots indicate significantly elevated mutation rates

To compare de novo mutations with natural variations, we merged the mutations from all MA lines and populations into 263 unique SNVs and 93 indels. We found that 64 (24%) of 263 total SNV sites coincide with biallelic SNPs in the 1001 Genomes dataset, and 50 (19% of the total) of these 64 shared SNVs are identical (Fig. 8). Among the 93 indels identified in MA experiments, 40 (43%) overlap with indels from the 1001 Genomes population, 12 (13% of the total) of which are identical. These identical sites (86% of shared SNVs, 75% of shared indels) are derived mainly from the Heat and Warming lines. Compared with the expected overlap (based on a random distribution of mutations and polymorphisms), the overlap between polymorphisms in all of our MA lines and populations with those of natural variants is highly significant (Fisher's exact test: SNV, $p = 2 \times 10^{-24}$; indel, $p = 1 \times 10^{-12}$; Fig. 8). To determine whether the SNVs identified in our MA results were biased toward conserved or substitution sites in *A. thaliana*, we compared them with the 219,909 ancestral variants (SNPs occurring at substitution sites in *A. thaliana*) and 1,799,125 derived variants (SNPs occurring at conserved sites) from the 1001 Genomes biallelic SNP dataset (see "Methods"). Among all SNVs identified in our MA results, only four SNVs (1% of SNVs from Warming C and F and Heat E) overlap with ancestral variants and one SNV (from Heat B) is shared with derived variants, indicating a low frequency of de novo SNVs (identified under elevated temperatures) at the conserved sites.

## Discussion
MA experiments in combination with whole-genome sequencing have been used to investigate spontaneous mutations in various model organisms with short generation times [1, 2, 14, 15, 23, 39, 40]. In unicellular MA studies, the organisms generally reproduce clonally (asexually), with each clonal population propagating for multiple generations, allowing the accumulation of mutations in a nearly neutral manner (minimal selection pressure) [1, 2]. By contrast, multicellular organisms usually produce progeny by sexual reproduction; thus, large numbers of progeny inevitably suffer from environmental selection pressure, resulting in accumulated mutations that differ from those arising from clonal reproduction. To avoid such selection effects, single-seed descent lines have been employed to accurately reflect mutation profiles under ideal conditions [23, 40]. In nature, however, plants generally occur in populations rather than in groups descended from a single seed. Therefore, a combination of MA populations and single-

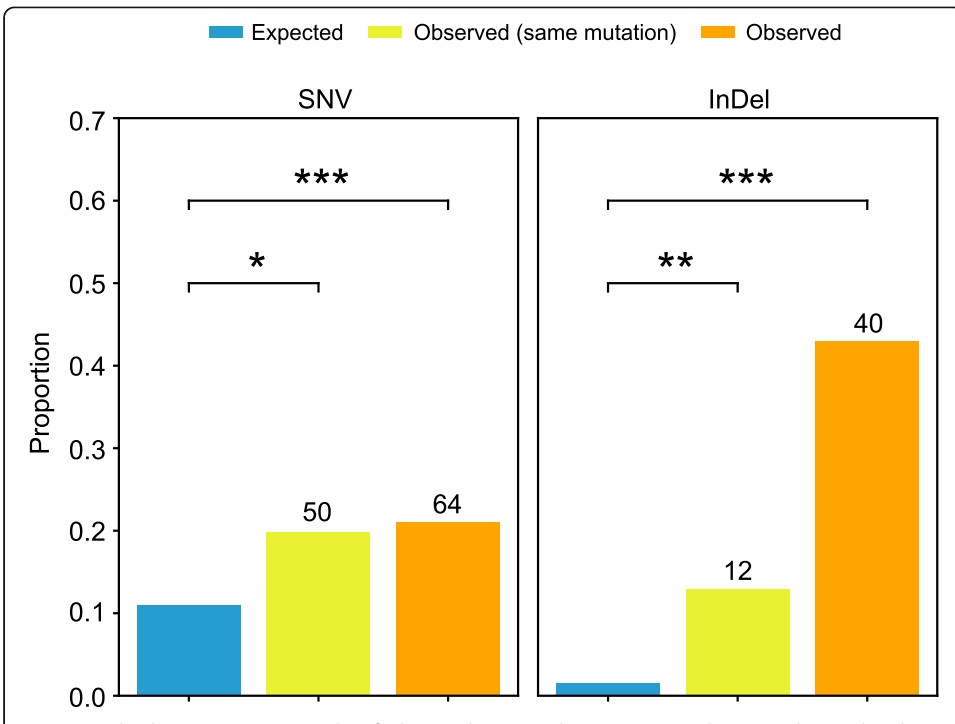

**Fig. 8** Overlap between mutations identified in MA lines/populations (SNVs) and variants detected in the 1001 Genomes population (SNPs). Comparison of expected and observed proportions of SNVs and indels that overlap the SNPs and indels in the 1001 Genomes dataset. Numbers at the tops of the bars are absolute overlap values. Asterisks indicate $p < 0.05$ (*), $p < 0.01$ (**), and $p < 0.001$ (***) based on Fisher's exact test with Bonferroni correction

seed descent MA lines could better reveal how spontaneous or environmentally induced MA shapes genetic variation. In our study, we employed both single-seed descent line and population MA experiments with *A. thaliana* grown under ambient, extreme heat, and moderately warming temperatures, and estimated their mutation profiles.

Environmental changes can promote genome-wide accumulation of mutations and change the profiles of new mutations in organisms, in turn potentially contributing to long-term evolutionary adaptation [8, 11, 24, 34, 41–44]. However, in plants, only one environmental factor (high-salinity soil) and artificial irradiation have been reported to induce MA [34, 45, 46]. In this study, we revealed that multigenerational exposure of *A. thaliana* to high and moderate heat could significantly increase the rates of accumulated de novo mutations in both MA lines and populations, which showed overall mutation rates that were more than 1.4-fold (1.49 for Warming C and 1.90 for Warming F) and 1.9-fold (1.91 for Heat B and 2.51 for Heat E) higher than Control groups. However, the rates of both spontaneous ($5.28$–$6.53 \times 10^{-9}$) and temperature-driven SNVs ($1.18 \times 10^{-8}$) in *A. thaliana* are lower than that of the long-lived perennial plants (*Populus trichocarpa*, $2.66 \times 10^{-8}$; Oak, $4.2$–$5.8 \times 10^{-8}$) [47, 48]. The majority of accumulated SNVs in trees were likely to be result from long-term exposure to ultraviolet light and longer lifespan (with higher number of mitoses), which is different from *A. thaliana* maintained in the laboratory (lower ultraviolet light and short life cycle, 2~3 months). On the other hand, the increased mutation rates under Heat and Warming temperatures were mainly attributed to the accumulation of a large proportion of SNVs

and small ($\leq$ 3 bp) deletions. This relatively high frequency of deletions at elevated temperatures provides evidence that *A. thaliana* genomes are still shrinking [49]. Moreover, these accumulated mutations in Heat (E, B) and Warming (C, F) treatments exhibited distinct molecular spectra compared to Control, with significantly increased rates of specific transversions (A:T $\rightarrow$ T:A) and transitions (C:G $\rightarrow$ T:A) under Heat treatment in particular. These results differ from those obtained in a salt stress MA study in which only the transversion rate was significantly increased [34], suggesting a possible divergence in response mechanisms driven by heat and salt stresses in *A. thaliana*. Nevertheless, the relatively low Ts/Tv ratios generated by Heat (1.56) and Warming lines (1.70) were similar to those generated by salt stress and irradiation in *A. thaliana* lineages [34, 45], further demonstrating that environment factors are responsible for changes in plant molecular spectra.

Given that altered lifespans, increased reactive oxygen species (ROS) levels (oxidative stress), and DNA repair impairments are induced by environmental stresses (including heat stress) in plants [26, 50–53], the increased mutation rates in *A. thaliana* grown under Heat and Warming temperatures may be explained by three previously proposed major causes of environmental stresses: generation time, metabolic rate, and DNA repair [26, 34, 54, 55].

Compared with MA lines, it is interesting that the accumulated mutations in MA populations showed different distribution pattern, e.g., some shared mutations retained in the final progeny. This result was probably due to the selective sweep or selection effects in MA populations, resulting in the relative concentration of mutations in some loci. Furthermore, our selection analysis on populations indicated that Heat MA population may have experienced a stronger positive selection effect (Ka/Ks > 1). In addition, the slightly lower rates of total mutations and coding mutations in population lineages than lines were likely also a result of our artificial selection procedure. A recent MA study of *C. elegans* showed that larger population sizes (N = 10, 100) with increasingly intense selection show lower spontaneous mutation rates compared to small (N = 1) populations [40]. This result, together with our results regarding the mutation rates of populations (N = 35) and lines (N = 1), may support the hypothesis that larger population sizes show lower mutation rates under selection pressure.

The datasets from 1001 Genomes provide abundant polymorphism resources for natural populations of *A. thaliana* [56]. The overlap between our identified de novo mutations and 1001 Genome biallelic variants was significantly greater than expected, indicating that these variants were not distributed uniformly in our MA results. When an organism is exposed to long-term environmental stress, the genomic distribution of accumulated variants usually changes compared to that under non-stress conditions, and such stress-induced mutations were increased in exonic regions of yeast and *E. coli*, for the purposes of evolutionary adaptation [8, 24]. In the present study, we observed significantly higher mutation frequencies and rates in coding regions of *A. thaliana* MA lines exposed to Heat (E) and Warming (F) temperatures compared to Control (D), especially nonsynonymous mutations; this suggests that elevated temperatures could accelerate the accumulation of mutations in coding regions. When the plants are exposed to high temperatures (especially in Heat MA lines), the accelerated rate of mutations in protein-coding regions may increase the load of deleterious mutations, thereby affecting fitness. Assuming that nonsynonymous SNVs (including stop codon

gain) and indels in coding regions affect fitness, the significantly higher mutation rates affecting fitness in the Heat and Warming treatments suggest that long-term exposure to high temperatures has a stronger effect on fitness and may contribute to genetic variation affecting fitness. In contrast to MA lines, the Heat MA population displayed lower mutation frequencies in coding regions but higher frequencies in noncoding regions, indicating that long-term heat stress coupled with strong selection constraints strongly restricted the accumulation of potentially deleterious mutations in *A. thaliana*. The low mutation rate affecting fitness estimated for the Heat MA population further suggests that the negative effects on fitness in this population may be offset by selective pressures, despite the increase in the genome-wide mutation rate.

Transposons are mobile DNA elements that are prevalent in eukaryotic genomes, and play important roles in regulating gene expression, accelerating sequence mutations, genome reshaping, and adaptive variation [57–61]. Transposon activity can be triggered by environmental stress, indicating that TEs can exhibit stress-responsive transposition or movement [62, 63]. In our MA study, we found higher mutation rates in TEs of the Heat and Warming groups than in those of the Control. Given that a high mutation rate in TEs could result in their transposable inactivation [64], we suggest that the growth of multiple generations at high temperatures can accelerate genome-wide transposon mutations in *A. thaliana*, consequently affecting stress-responsive transposition or movement of TEs.

Methylated cytosines contribute to changes in spontaneous deamination to thymidine, thereby leading to elevated mutation rates in MA lines [23, 37]. Consistent with these findings, the present study revealed that the frequency of methylation (in all three contexts) at mutated bases is significantly elevated in plants exposed to high temperatures. Further analysis confirmed that methylated sequences were associated positively with mutations in the Heat and Warming MA lines. These results suggest that methylation contributes to the accumulation of mutations at high temperatures. In addition, the occurrence of DNA methylation was biased toward TE regions [35, 38]. Indeed, we observed higher mutation rates of methylated sites within TEs, suggesting a universal mechanism of methylated TE-biased mutations in *A. thaliana* regardless of environmental factors. However, heat treatment increased the mutation rate at methylated sites outside TEs, suggesting that methylation within and outside of TEs contributes to the high mutation rate observed after multigenerational exposure of *A. thaliana* to heat stress.

Plants subjected to excess heat exhibit a characteristic set of cellular and metabolic responses, and transcription responses [26, 65, 66]. In this study, we observed that mutigenerational growth of *A. thaliana* at high temperatures led to greater accumulation of mutations in genes involved in responses to stimuli, stress response (*AT5G48770* and *AT4G10780*), reproduction, replication and repair, and development, relative to growth under control conditions. These results suggest that mutigenerational growth of *A. thaliana* at high temperatures causes mutations in temperature response- or stress-associated genes. Given that mutations in some functional genes (e.g., loss-of-function) are important for plant adaptation [67, 68], whether these gene mutations contribute to high-temperature adaptation of plants requires further investigation. We also found that one gene (*AT4G17760*) associated with DNA repair and DNA damage checkpoints was mutated in a Heat MA line. Under heat stress, ROS-induced DNA

damage is usually repaired by both the nucleotide and base excision repair processes, during which DNA repair pathways are activated [52, 69]. These results suggest that multigenerational exposure of *A. thaliana* to high temperatures promotes the mutations in stress response genes, thereby exerting influences on physiological and developmental processes in response to long-term high-temperature stress.

Elevated temperatures, including both extreme heat and moderate warming, have reportedly caused genetic variations in plant phenology and developmental timings [26, 33, 70–73]. However, how plants respond to long-term moderate or extreme heat stress at the whole-genome level remains unknown. Our results from both MA lines and populations revealed higher mutation rates of accumulated SNVs and indels in *A. thaliana* exposed to multigenerational heat stress (Heat E, B) compared to moderate warming (Warming F, C). Moreover, almost all mutation types in both genic and intergenic regions showed higher mutation frequencies under Heat compared to Warming temperatures. The lower Ts/Tv ratio in Heat E (1.56) compared to Warming F (1.70) suggests that plants exposed to extremely high temperatures might accumulate more deleterious variations (e.g., more transversions in coding regions). Possible explanations for this include greater DNA damage driven by increased amounts of free radicals under environment stress [34, 54], and successive induction of DNA damage due to long-term heat stress coupled with greater impairment in DNA repair mechanisms. In addition to these differences, a common mutation bias toward a high rate of A:T → T:A (transversions) was observed in the Heat and Warming groups, which suggests potential convergence of the mutation mechanisms induced by the two elevated temperature treatments.

The effect of epigenetic variations (e.g., DNA methylation) on the probability of mutation may be ubiquitous in plants, and more epigenetic modifications occur in response to environmental stresses [23, 34, 74–76]. Thus, investigations into epigenetic variations, and in particular epimutations across generations (non-transient variations), could shed additional light on the genetic evolutionary mechanisms activated in response to long-term high-temperature stress.

## Conclusion

Growth of *Arabidopsis* under elevated temperatures over multiple generations significantly accelerated mutation accumulation and altered mutation profiles. The increased mutation frequency occurred in intergenic regions, coding regions, and transposable elements, as well as in nonsynonymous mutations in functional genes. Different mutation distribution patterns of progeny between populations and lines suggested stronger selection effects on populations. Our results reveal the fundamental mutation spectrum under environmental warming, improving the understanding of genetic variations in plants under long-term heat stress.

## Methods

### Plant materials, experimental design, temperature treatments, and growth conditions for the MA experiments

We used the wild-type Columbia (Col-0) strain of *A. thaliana* to conduct MA experiments; this strain has been maintained for several years in our laboratory [53]. A single

ancestral Col-0 plant was planted in soil and grown in a growth chamber under a 16-h light/8-h dark photoperiod and 150 μmol m$^{-2}$ s$^{-1}$ of photosynthetically active radiation (PAR) provided by fluorescent tubes (Philips Electronics Trading & Services, Shanghai, China), at 23/18 °C (light/dark) ambient temperature and 50/80% relative humidity (RH). In addition, the plant was alternately watered to saturation with 1/2 Murashige and Skoog (MS) solution or deionized water every week. The ancestral plant was self-pollinated, and its progeny seeds were harvested for our MA line and population experiments. The MA line and population experiments were performed under 23/18 °C (light/dark), 28/23 °C, and 32/27 °C temperature regimes, and all detailed treatment methods are described in the Additional file 3: Supplementary detailed methods.

### DNA extraction, sequencing, quality trimming, and filtering

The 15 MA lines (D10, E10, F10) at G10, and 15 plants from MA populations (five plants per population) at G16 (Control, A16), G22 (Heat, B22), and G19 (Warming, B19) (Fig. 1a), were chosen for sampling and sequencing. When each plant was grown to the bolting stage [leaves were fully expanded and matured in this stage; Fig. 1b (growth stage 5 according to [77])], the leaves were sampled and immediately frozen in liquid nitrogen. Similarly, the leaves of five plants representing the progenitor were also collected separately. Genomic DNA was extracted and purified from each plant using a MiniBEST Plant Genomic DNA Extraction Kit (TaKaRa, Dalian, China). To obtain DNA samples representing the progenitor (G0) (Fig. 1a), genomic DNA samples from five independent plants (G0-1–G0-5) were used to construct five separate libraries. These five libraries were combined to constitute the genetic background of the progenitor. All DNA sequencing (paired-end) projects were performed by Frasergan Genomics Institute (Wuhan, China) and Novogene (Beijing, China) according to the manufacturer's specifications (Illumina, San Diego, CA, USA). Finally, the Illumina HiSeq 2500 and HiSeq X Ten platforms were employed to generate raw sequences with 125-bp (MA populations and G0) and 150-bp (MA lines) read lengths, respectively.

Raw reads were cleaned by removing the following types of reads: (1) reads containing adaptor sequences; (2) reads with Phred quality ≤ 20; and (3) reads with length ≤ 50 bp. The remaining paired-end reads were used for variant calling. Consequently, we obtained approximately 190 Gb of high-quality sequences with an average of 43.3× genome coverage, which were used for subsequent analyses (Additional file 1: Table S1).

### Sequence alignment, variant calling, and annotation

We aligned the high-quality reads from each sample to the *A. thaliana* genome (TAIR10) using the BWA-MEM (version 0.7.16a) algorithm [78]. To obtain high-quality SNVs and indels from each sample, we applied the Genome Analysis Toolkit (GATK, version 4.0.4.0) [79] procedures to perform SNV and indel callings. In the recommended GATK procedure, after removing the duplicate reads with Picard MarkDuplicates, we called the SNVs and indels using the HaplotypeCaller method [80]. Subsequently, the variant calls from GATK were further filtered with the following parameters: (1) for SNVs, QD < 2.0 || FS > 60.0 || MQ < 20.0 || MQRankSum < − 12.5 || ReadPosRankSum < −8.0, DP < 8; and (2) for indels, QD < 2.0 || FS > 200.0 || MQ < 20.0 || ReadPosRankSum < − 20.0, DP < 5. After filtering, the preliminary high-quality

SNVs and indels from each sample were retained for base quality score recalibration (BQSR), which was performed using the BAM files generated through the GATK calling method described above.

To ensure the accuracy of de novo mutations called in the MA lines (G10) and populations (G16 for Control, G22 for Heat, G19 for Warming), we employed the following stringent criteria: (1) for MA and ancestor (background) samples, SNVs and indels must be called in at least eight reads and five reads, respectively; (2) the called SNVs must include both the forward and reverse reads; (3) in each MA sample, the mutation candidates were compared to the background (G0-1–G0-5) to discard spurious candidates resulting from variation between the reference genome sequence and the genome of the background; (4) owing to alignment difficulties in the vicinity of indels, SNVs located around indels (< 10 bp on each side) in each MA sample were removed; (5) called indels with a ≤ 20-bp interval between them were discarded; (6) called indels that were repeated multiple times at that site in the reference genome were subjected to further validation through Sanger sequencing to ensure the accuracy of the indels. In addition, we further filtered the variant sites (SNVs and indels) that were heterozygous in G0 samples, resulting in the identification of homozygous SNVs and indels in each MA sample. Finally, all filtered variants remaining in the MA lines and populations were inspected manually using Integrated Genome Viewer software [81]. Accordingly, we obtained a list of reliable new homozygous variants in our MA lines and populations. De novo TE insertions in each sample were detected with the Jitterbug software package [82]. There were three main steps in this process. First, the deduplicated alignment BAM files (for each MA sample) and TE annotation (from TAIR10) were used as the input for Jitterbug. All parameters were kept as default with the exception of "--mem" parameter. Second, high-quality predictions were selected using jitterbug_filter_results_func.py under default settings as described in https://github.com/elzbth/jitterbug. Third, the predicted results of TE for each sample from the MA lines and the MA populations were further filtered by removing TE predictions found in the progenitor samples using BEDTools v2.29.2 [83].

To validate the reliability of our mutation calling pipeline, we conducted two types of simulation. For the first simulation, 600 random point mutations from throughout the reference genomes of six selected MA lines and populations (with different reference genome coverage depths, 100 mutations per reference) were simulated to test whether our pipeline could recover them [14, 35]. The simulated mutation positions remained independent among the six reference genomes. Twenty mutations per chromosome were simulated. Each clean read from these MA lines and populations was mapped separately to one of the mutated reference genomes, and then their genotypes were called using GATK. The number of simulated mutations recovered and the corresponding accessible nonvariant sites were reported. For the second simulation, homozygous mutations were introduced to the sequencing reads in the BAM files for the same MA lines and populations; for detailed simulation methods, refer to [35].

We then used the ANNOVAR software [84] to annotate the identified SNVs and indels and divided them into groups of variations occurring in intergenic regions, exonic regions (overlapping a coding exon), intronic regions (overlapping an intron), splicing sites, upstream or downstream regions (within a 1-kb region upstream from the transcription start site or downstream from the transcription stop site), UTR regions,

noncoding RNAs, and pseudogenes, on the basis of *A. thaliana* genome annotation information (TAIR10). SNVs in coding regions were further divided into nonsynonymous SNVs (those that caused amino acid changes) or synonymous SNVs (those that did not cause amino acid changes); frameshift, stop-gains, and stop-losses caused by mutations were also classified in coding regions. In addition, an Araport11 genes and transposons GFF file [85] was used to further annotate mutations occurring in TEs. In addition, previous studies have suggested that 66% of mutations in coding regions could be purged through natural selection. According to the method of [23], we estimated the genomic mutation rate (diploid) affecting the fitness of each MA line by multiplying our mutation rate (per site per generation) in the line's coding region by the mean selective constraint of 0.66, and then doubling that value to obtain the total number of coding sites. Finally, the average genomic mutation rates affecting fitness in the Control, Heat, and Warming groups were calculated. In addition, the estimation of Ka/Ks ratios were conducted using KaKs calculator software (version 2.0, [86]).

To identify the mutations occurring in potential temperature-responsive transcripts in MA lines and population, we firstly retrieved the previous transcriptome datasets (deposited in GSE118298 [72];) of *A. thaliana* leaves grown at prolonged warming, heat shock, and control. Then, differentially expressed genes (control *vs* heat shock; control *vs* prolonged warming) occurred in the mutated gene of MA lines and populations were considered as temperature-responsive transcripts.

### Sanger sequencing validation

We employed Sanger sequencing to estimate the accuracy of our mutation calls. All identified mutation calls (211 mutations) including SNVs and indels from MA lines were selected for verification. Primers were designed around the mutations using Primer5 to amplify a 200–600-bp region (Additional file 1: Table S6). All but six mutations were verified through PCR as being present in MA lines but absent in the progenitor (background, G0), indicating a negligible false-positive detection rate; six mutations could not be verified through PCR amplification.

### Mutation rate calculations

We calculated the mutation frequency (per genome per generation) and rate (per site per generation) using the formulas $m = n/g$ and $\mu = n/gb$ (or $m/b$), respectively, where $n$ is the total number of mutations per line detected, $g$ is the number of generations, and $b$ is the number of bases (*A. thaliana* genome bases) analyzed. Detailed methods are illustrated in Additional file 3: Supplementary detailed methods. In addition, we also calculated the conditional mutation rates for all six substitution possibilities in each line and population using the same equations, but with $m$ representing the number of substitutions of the focal substitution type, and $n$ representing the total number of analyzed sites capable of producing a given mutation. For our analysis of mutation rates within chromosomes with different regions, we first divided each chromosome into multiple 100-kb bins, and then classified these bins into high and low gene density groups based on the gene distribution in each chromosome. Mutation rates in each group were measured by dividing the total number of SNVs and indels from our study (Control, Heat, and Warming) by the product of the total number of sites in each

group and the number of generations per line using the following formula: $\mu = n/gb$. Although the actual mutation rate in each treatment might be underestimated, this has little effect on the mutation rate comparisons (see Additional file 3: Supplementary detailed methods).

### Comparison of genetic variations between MA and natural populations

All VCF files of natural variations were downloaded from the 1001 Genome Project website (http://1001genomes.org). SNP loci and allele frequencies were assessed. Ancestral and derived variants were defined according to Weng et al. [35]. Briefly, triple whole-genome alignment among *A. thaliana* (TAIR10), *A. lyrata*, and *Capsella rubella* was performed, and identical sites within the alignment regions of all three species were identified as conserved. Sites that were the same in *A. lyrata* and *C. rubella*, but different in *A. thaliana* were considered to be ancestral. De novo variants from MA lines or populations at conserved and ancestral sites were identified as derived and ancestral mutations, respectively.

### DNA methylation and interaction analysis

Whole-genome bisulfite sequencing (WGBS) was conducted using the same genomic DNA extracts used for whole-genome sequencing. Genomic DNA was fragmented to 200–300 bp with a Covaris sonicator, followed by end repair and adenylation. The purified DNA fragments were bisulfite converted using the EZ DNA Methylation-Gold$^{TM}$ Kit (Zymo Research). WGBS libraries were sequenced on the Illumina HiSeq 2500 platform (Novogene, China). The raw paired reads were quality controlled and trimmed, and then aligned to the reference genome (TAIR10) using Bismark software v0.16.1 [87] with the default parameters. Methylated cytosines (mCs) in each sequence were extracted from the aligned reads using the Bismark methylation extractor. The proportions of mCG, mCHG, and mCHH (where H refers to A, T, or G) contexts were calculated by dividing the number of mCs by the total number of Cs. Methylation-level analysis was performed using divided bins (bin size = 100 bp). The methylation level (ML) of each C was defined as reads (mC) / reads (mC) + reads (C). mCs with corrected $p$ values ≤ 0.05 were used for subsequent interaction analysis. A logistic regression model was used to test the effects of C methylation and TE on the mutation probability of a given nucleotide. Analysis of interaction among cytosine methylation (mC or nonmethylated cytosine), and TE (TE or non-TE) was generally conducted according to the method of Weng et al. [35].

### Local sequence context and GC content analyses

To investigate the effect of local sequence context on the mutation rate in each treatment, we extracted the flanking bases (1 bp upstream and downstream of the mutation site) at A/T and G/C positions surrounding the mutation sites. We calculated the rates of trinucleotide mutation in MA lines and populations under the Control, Heat, and Warming conditions. In addition, we estimated the relative contribution of the GC content to mutability through calculating the mutation rate under a given GC content, using a sliding window of 1000 bp (bin size = 1000 bp) and 0.005 intervals of GC content. The formula used is $\mu = n / (m \cdot g \cdot 1000)$, where $\mu$ represents the mutation rate

under a given GC content, $m$ represents the number of bins in a given GC content, and $n$ represents the total number of mutations within $m$.

## Statistical analyses

All statistical analyses were performed in R Studio (version 1.0.143) using the Stats analysis package (R Development Core Team, Vienna, Austria) and SPSS software (version 23.0; IBM Corp., Armonk, NY, USA). $p$ values $< 0.05$ and $< 0.01$ were indicative of significant and very significant results, respectively.

## Supplementary Information

---

**Additional file 1: Table S1**. Summary of whole-genome sequencing datasets of DNA samples. **Table S2**. Number of de novo mutations identified from MA lines and MA populations grown under Control, Heat, and Warming conditions. **Table S3**. List of individual sequence mutations identified in Control, Heat, and Warming MA lines and populations. **Table S4**. Simulated reference mutations recovered from six randomly selected MA lines. **Table S5**. Simulated sequence read mutations recovered from six randomly selected MA lines. **Table S6**. Validation of mutations detected in MA lines through conventional Sanger sequencing. **Table S7**. Numbers and frequencies of transitions (ts) and transversions (tv) identified in MA lines and populations grown under Control, Heat, and Warming conditions. **Table S8**. Comparison of mutation frequency among different genic regions calculated from Control, Heat, and Warming MA lines (A) and MA populations (B). **Table S9**. Whole-genome bisulfite sequencing and the numbers of methylated cytosine positions in genomic DNA samples. **Table S10**. Logistic regression analysis of the effects of cytosine methylation and TE regions on the likelihood of a given nucleotide being mutated in the Control, Heat, and Warming MA lines. **Table S11**. Genomic locations and affected bases of indels under Control, Heat, and Warming conditions.

**Additional file 2: Fig. S1**. KEGG enrichment of mutated genes in the Control (A and D), Heat (B and E), and Warming (C and F) treatments. **Fig. S2**. Mutation rates estimated based on the interactions between TEs/non-TEs and methylation/unmethylation sites in the Control D (A), Heat E (B) and Warming F (C) MA lines. The interactions were classified as follows: non-TE with unmethylated sites, non-TEs with methylated sites, TEs with unmethylated sites, and TEs with methylated sites. In each case, the mutation rate was calculated by dividing the number of observed mutations by the number of analyzed sites capable of producing a given mutation, and the number of generations of MA in each Control, Warming, and Heat line. Differences in mutation rates were evaluated using Student's $t$-test. Asterisks indicate significant differences from Control D at $p < 0.05$ (*). **Fig. S3**. Relative mutation rates by local GC content across the *A. thaliana* genome of MA lines (Control D, Heat E, and Warming F) and populations (Control A, Heat B, and Warming C). A 1-kb bin size and 0.005 intervals of GC content were used (see Methods). The figure was plotted using the loess method and stat_smooth function of the *ggplot2* R package. Relative mutation rate was calculated as described in Methods.

**Additional file 3: Supplementary detailed methods**.

**Additional file 4.** Review history.

---

### Acknowledgements
We thank Yang Wang, Jing Wang, Xiaoxue Jiang, Ming Fang, Kaige Luo, Kaibiao Ma, Zhichao Jia, and Shixiong Ren for contributing to maintaining the mutigenerational growth of materials over the past 10 years. We also thank Frasergen and Novogene companies for assistance with genome sequencing and technical support. No conflict of interest declared.

### Peer review information

### Review history
The review history is available as Additional file 4.

### Authors' contributions
B.J., C.X., L.J, and N.T. conceived and designed the project. Z.L. and J.C. performed mutigenerational growth of materials and seed harvest, and prepared the samples for sequencing. Z.L., S.X., and W.K. designed the analytical approach and performed data analysis. Z.L. and J.C. performed the PCR validations. Z.L., L.W., and B.J. wrote the manuscript. Y.Z., S.Z., and H.-M.L. contributed to conceptual and analytical inputs of the paper. The authors read and approved the final manuscript.

### Funding
This work was supported by the National Natural Science Foundation of China (grant no. 30870436), the Hong Kong Research Grants Council Area of Excellence Scheme (AoE/M-403/16), the Program of Introducing Talents of Discipline to Universities (111 project, B13007), and self-raised funds.

### Availability of data and materials

All whole-genome sequencing data from this study have been deposited in the NCBI under the accession number PRJNA548479 (https://www.ncbi.nlm.nih.gov/bioproject/PRJNA548479) [88]. The whole-genome bisulfite sequencing datasets of MA lines have been deposited in NCBI Gene Expression Omnibus with the accession number GSE173660 (https://www.ncbi.nlm.nih.gov/geo/query/acc.cgi?acc = GSE173660) [89].

## Declarations

### Ethics approval and consent to participate

Not applicable.

### Consent for publication

Not applicable.

### Competing interests

The authors declare that they have no competing interests.

### Author details

[1]College of Horticulture and Plant Protection, Yangzhou University, Yangzhou, China. [2]Key Laboratory of Plant Functional Genomics of the Ministry of Education, Agricultural College of Yangzhou University, Yangzhou, China. [3]College of Horticulture, Nanjing Agricultural University, Nanjing, China. [4]Center for Soybean Research of the State Key Laboratory of Agrobiotechnology and School of Life Sciences, The Chinese University of Hong Kong, Shatin, Hong Kong Special Administrative Region, China. [5]Institute of Plant Stress Biology, State Key Laboratory of Cotton Biology, Department of Biology, Henan University, Kaifeng, China. [6]Wuhan Frasergen Bioinformatics Co, Wuhan, China. [7]College of Biological Sciences and Technology, Beijing Forestry University, Beijing, China.

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

## 