## [**Additional file 4.** Review history. · Genome Biology]

Review History

First round of review

Reviewer 1

Are you able to assess all statistics in the manuscript, including the appropriateness of statistical tests used? No, I do not feel adequately qualified to assess the statistics.

Comments to author:

This manuscript studied genome evolution of the accumulated lines or populations of Arabidopsis in high temperature. Basically, this is a well-designed study, and most analyses are very well.

However, I do have some concerns and questions:

1) In the abstract, authors claimed "Mutation occurred more frequently in intergenic regions, coding regions (especially nonsynonymous mutations), and transposable elements (TEs)", but in the later section, authors mentioned "the mutations in lines and populations grown under elevated temperatures were significantly biased toward low gene density regions, special trinucleotides (GC context), tandem repeats, and adjacent simple repeats". These two sentences are kindly contradict to each other.

2) No any novel TE insertion in any MA line or population, however, it is not clear how authors searched the new TEs across the genome in the present manuscript.

3) Apparently, authors presents results MA lines first in the later Figures, it is better to exchange the positions of MA lines with MA population in Figure 1a.

4) On Figure 5c, the color key of frameshift insertion and nonsynonymous is very similar, please change the color make it easy to be differentiated from each other.

5) For the head of the each section in the results, please directly replace plants with Arabidopsis.

6) Figure legends is not clear enough, for example, figure legend for Figure 4, the mutation frequency one would expect the value would be 1, but here the value is larger than 1, more detailed explanation is needed.

7) From line 258 to line 263, the SNV rate of genic region in Heat B is higher than that in warming C, instead of lower as authors wrote. I guess here authors want to say mutation frequency of coding in Heat B is lower than that in warming C. Furthermore, it is not clear why the pattern of mutation frequency of coding in MA lines (larger in Heat B) and MA populations (lower in Heat B) is different. More comparison should be performed in the whole ms between MA lines and MA populations.

8) From line 269 to 270, the sentence "In the comparison among lines, lower mutation rates affecting fitness were found in MA populations.", it is a little ambiguous, maybe remove it directly.

9) Line 351-353, please rephrase this sentence somehow, it is the result of evolution.

10) The discussion section is too longer, only discuss the most important results is enough, instead of repeat or emphasize every results one by one.

Reviewer 2

Are you able to assess all statistics in the manuscript, including the appropriateness of statistical tests used? Yes, and I have assessed the statistics in my report.

Comments to author:

In the submitted manuscript Lu et al. estimated Arabidopsis mutation rate under elevated temperature in multi-generation mutation accumulation lines (single seed descent) and mutation accumulation populations (35 individuals). Their results showed the mutation rate were elevated when plants were grown under high temperature conditions. The estimated mutation rate, mutational spectrum, exceeding overlaps between spontaneous mutation and natural variants were all consistent with previous studies. The manuscript was well-written and the authors explained the experimental protocol and variant calling procedure in details.

The study confirmed several aspects of mutation parameters in Arabidopsis previously shown. One uniqueness of the study was the MA population. I felt the comparison between MA population and MA line was underutilized in the manuscript that might inform us about the process of selection, fitness, population size and mutation. For example, comparing to MA lines (n=1), the decrease of mutation rate at coding regions in MA population (n=35) might be utilized to estimate the selection strength and the fitness effects on these coding genes.

Under heat, the authors found mutations at DNA repair genes and also high mutation rates. These two observations could be connected and have nothing to do with heat. I wonder if the authors have considered a different scenario. Mutation occurred at DNA repair DNA just by chance in the heat MA line, and has nothing to do with the heat treatment. Because DNA repair gene was mutated, the mutation rate also increased. So the observed the mutation rate increases in heat was due to the mutation in DNA repair gene rather than heat. Please comment on this.

Minor comments:

1. Line 78. The authors define large duplications/deletions as del, but this is not consistent with line 161 where del refers to 1-3 short deletion. Please make changes and be consistent.
2. Line 181-205 and figure 3. I would suggest including the standard error of the mean (SEM) in writing. SEM was shown in figure 3 but not written in line 181-205.
3. Line 352, 377 and lines 553-555. The authors used the word "ability" to describe the bias in mutation toward low gene density region and certain trinucleotide context. I found it a bit awkward to describe plants can control their mutation in a certain way. Changes in mutational parameters should be considered as an outcome the interplays between multiple evolutionary processes rather than controlled by the plant itself. I would suggest to modify the wording.

4. Figure S2 has a mismatched figure legend.
5. Figure S3. It's not clear why the non-linear smooth function was used in the figure.
6. Additional file 3. Page 6. The citation for GSE118298 was incorrect.

Reply to the editor

Previous manuscript ID: GBIO-D-20-01808

Title: Genome-wide DNA mutations in *Arabidopsis* plants after multigenerational exposure to high temperature

Authors: Zhaogeng Lu, Jiawen Cui, Li Wang, Nianjun Teng, Shoudong Zhang, Hon-Ming Lam, Yingfang Zhu, Siwei Xiao, Wensi Ke, Jinxing Lin, Chenwu Xu, Biao Jin

Dear Dr. Pang,

We thank you for your kind decision and the reviewers for their valuable comments and suggestions. The issues raised by the reviewers were of great help for our revision. In the past few weeks, we have made great efforts to revise the original manuscript. In the revised manuscript, all the questions and suggestions were taken into consideration and the revised parts were marked in blue. Below we provided a point-to-point response.

Response to the reviewer I

Comments to the Author:

This manuscript studied genome evolution of the accumulated lines or populations of *Arabidopsis* in high temperature. Basically, this is a well-designed study, and most analyses are very well. However, I do have some concerns and questions:

Response: We thank you for carefully reviewing our manuscript and putting forward many valuable comments and suggestions. We have revised our manuscript according to the comments and suggestions by you and another reviewer.

Question 1: In the abstract, authors claimed "Mutation occurred more frequently in intergenic regions, coding regions (especially nonsynonymous mutations), and transposable elements (TEs)", but in the later section, authors mentioned "the mutations in lines and populations grown under elevated temperatures were significantly biased toward low gene density regions, special trinucleotides (GC context), tandem repeats, and adjacent simple repeats". These two sentences are kindly contradict to each other.

Response: Thank you for pointing out this issue. After reading your comment, we realized that the two sentences seem to be confusing. We originally tried to emphasize the two different mutation properties in the lines and populations grown under elevated temperatures.

For the first sentence "Mutation occurred more frequently in intergenic regions, coding regions (especially nonsynonymous mutations), and transposable elements (TEs)", we actually meant the relatively higher mutation frequency in intergenic, coding regions and TEs of plants grown under elevated temperatures **compared to those under control temperature.**

For the second sentence "the mutations in lines and populations grown under elevated temperatures were significantly biased toward low gene density regions, special trinucleotides (GC context), tandem repeats, and adjacent simple repeats", we were comparing the mutation rates among different genomic regions **within the DNA** in the plants obtained under elevated temperatures, but **not a comparison between plants under elevated temperatures and control.**

We have revised the two sentences into the followings:

"Mutation occurred more frequently in intergenic regions, coding regions, and transposable elements in plants grown under elevated temperatures than in control plants."

"Mutations occurring within the same genome under elevated temperatures were significantly biased toward low-gene-density regions, special trinucleotides (GC context), tandem repeats, and adjacent simple repeats."

Question 2: No any novel TE insertion in any MA line or population, however, it is not clear how authors searched the new TEs across the genome in the present manuscript.

Response: De novo TE detection for each sequenced genome (each sample) in our study was performed using Jitterbug (Henaff et al., 2015, BMC Genomics). There were three main steps in this process. First, the deduplicated alignment BAM files (for each MA sample) and TE annotation (from TAIR10) were used as the input for Jitterbug. All parameters were kept as default with the exception of "--mem" parameter. Second, high-quality predictions were selected using jitterbug_filter_results_func.py under default settings as described in <https://github.com/elzbth/jitterbug>. Third, the predicted results of TE for each sample from the MA lines and the MA populations were further filtered by removing TE predictions found in the progenitor samples using BEDTools v2.29.2 (Quinlan et al., 2010, Bioinformatics). Based on above methods, we did not find any novel TE insertion in our MA lines and populations.

We have added the detection methods of novel TE insertion in the Supplemental methods (Please see Additional file 3).

References:

Hénaff E, Zapata L, Casacuberta JM, Ossowski S. Jitterbug: somatic and germline transposon insertion detection at single-nucleotide resolution. BMC Genomics. 2015, 16:768. doi: 10.1186/s12864-015-1975-5.

Quinlan AR, Hall IM. BEDTools: a flexible suite of utilities for comparing genomic features. Bioinformatics. 2010, 26(6):841-842. doi: 10.1093/bioinformatics/btq033.

Question 3: Apparently, authors presents results MA lines first in the later Figures, it is better to exchange the positions of MA lines with MA population in Figure 1a.

Response: As suggested, we have exchanged the positions of MA lines with MA populations in Figure 1a in the revised manuscript. For consistency, we also exchanged the positions of MA lines with MA population in Figure 1b.

Question 4: On Figure 5c, the color key of frameshift insertion and nonsynonymous is very similar, please change the color make it easy to be differentiated from each other.

Response: Thank you for your suggestion. We have changed the color key of frameshift insertion in Figure 5c in the revised manuscript.

Question 5: For the head of the each section in the results, please directly replace plants with Arabidopsis.

Response: We have replaced “plants” with “*Arabidopsis thaliana*” in the head of each section in the Result section.

Question 6: Figure legends is not clear enough, for example, figure legend for Figure 4, the mutation frequency one would expect the value would be 1, but here the value is larger than 1, more detailed explanation is needed.

Response: According to your suggestion, we have added more details to the legends of some figures, including Figure 4, Figure 5, Figure S2, and Figure S3.

For Figure 4, the mutation frequency (m) of each region (for each sample) was calculated using the formulas $m = n/g$, where n is the number of identified mutations and g is the number of generations. Accordingly, the mean mutation frequency of each treatment (five samples) was estimated by the $\sum m/5$. Therefore, the numerical values (in the bar, for example, 0.68, 0.34, 0.24 and 0.20 in Fig. 4a) represent the mutation frequency of each genomic region, and the total mutation frequency of four genomic regions is not equal to “one”. Consequently, the values do not represent proportions of four genomic regions.

In addition, the SNV rates of each genomic region (per site per generation, the numerical values above bars of Fig. 4) were estimated by dividing the number of observed mutations by the number of analyzed sites capable of producing a given

mutation and the number of generations of MA. These estimation methods of mutation frequency and mutation rate were also included in our supplemental methods in Additional file 3 (“Mutation rate estimation”).

Question 7: From line 258 to line 263, the SNV rate of genic region in Heat B is higher than that in warming C, instead of lower as authors wrote. I guess here authors want to say mutation frequency of coding in Heat B is lower than that in warming C. Furthermore, it is not clear why the pattern of mutation frequency of coding in MA lines (larger in Heat E) and MA populations (lower in Heat B) is different. More comparison should be performed in the whole ms between MA lines and MA populations.

Response: Thank you for pointing out the issue and your valuable comments.

1) We are sorry that the original sentence was not described correctly. We meant that the mutation frequency of coding region in Heat B is lower than that in warming C. We have corrected the description in the revised manuscript.

2) In regard to “why the pattern of mutation frequency of coding in MA lines (larger in Heat E) and MA populations (lower in Heat B) is different”, we have carefully analyzed the data and the reasons are as the followings: i) our population experiment was designed to simulate the mutation accumulation in natural condition, whereas plants under natural condition generally occur in populations rather than single-seed descendants over multiple generations; ii) the large bottleneck (from 35 seedlings to 5 seedlings) in MA population would inevitably result in an increase of selection pressure, consequently affecting the mutation accumulation (or genetic drifting) in MA population; iii) in contrast, the MA lines were propagated in each generation only from five independent seedlings, which minimize the selection effect on the mutation accumulation during propagation. Therefore, when plants under mutigenerational exposure to heat, the larger selection pressure (or selective sweep) in MA population would decrease the accumulation of deleterious mutations, whereas the lower selection effect on mutation accumulation of MA lines would be negligible. In addition, our

supplemental results of fitness estimation also suggest that genomic mutation rates affecting fitness in Heat B and Warming C populations is lower than those of Heat E and Warming F MA lines.

All these reasons above explain why the pattern of mutation frequency of coding regions in Heat E lines is higher than Heat B population.

3) According to your suggestions, we have performed more comparisons between MA lines and MA populations at elevated temperatures, including the frequencies of transition and transversion, coding regions, nonsynonymous mutations, and the estimation of genomic mutation rate affecting fitness. Moreover, based on the occurrence of mutations in coding regions, we further conducted the estimation of selection pressure on coding genes between MA lines and MA populations using KaKs calculator. The Heat lines exhibited a Ka/Ks ratio of 0.92, whereas Heat population showed a Ka/Ks ratio > 1 (1.51). This result supports our previous speculation that Heat population probably have subjected to stronger positive selection than MA lines.

In addition, we also discuss the different distribution patterns of mutations among all individuals between MA populations and MA lines. In each MA population, some mutations are shared by different individuals and retained in the final progeny; whereas the mutations accumulated in MA lines were diversely scattered in the individuals (no common mutations). This striking result was probably due to the selective sweep (or selection effects) in MA populations, resulting in the relative concentration of mutations in some genomic loci.

All these comparative results and analyses have been included in the corresponding Abstract, Results, and Discussion sections of the revised manuscript.

Question 8: From line 269 to 270, the sentence "In the comparison among lines, lower mutation rates affecting fitness were found in MA populations.", it is a little ambiguous, maybe remove it directly.

Response: As suggested, we have deleted the sentence in the revised manuscript.

Question 9: Line 351-353, please rephrase this sentence somehow, it is the result of evolution.

Response: As suggested, we have rephrased this sentence into “This result suggests that multigenerational exposure of *A. thaliana* to high temperatures accelerates the accumulation of DNA mutations toward low gene density region compared to plants under ambient (control) temperature”.

Question 10: The discussion section is too longer, only discuss the most important results is enough, instead of repeat or emphasize every results one by one.

Response: We have simplified the Discussion section and emphasize the important results in the revised manuscript. Please see page 25, lines 520, 523-525; page 26, lines 544-545, 546-547, 550; page 27, line 565.

Response to the reviewer II

Comments to the Author:

In the submitted manuscript Lu et al. estimated Arabidopsis mutation rate under elevated temperature in multi-generation mutation accumulation lines (single seed descent) and mutation accumulation populations (35 individuals). Their results showed the mutation rate were elevated when plants were grown under high temperature conditions. The estimated mutation rate, mutational spectrum, exceeding overlaps between spontaneous mutation and natural variants were all consistent with previous studies. The manuscript was well-written and the authors explained the experimental protocol and variant calling procedure in details.

The study confirmed several aspects of mutation parameters in Arabidopsis previously shown. One uniqueness of the study was the MA population. I felt the comparison between MA population and MA line was underutilized in the manuscript that might inform us about the process of selection, fitness, population size and mutation. For example, comparing to MA lines (n=1), the decrease of mutation rate at coding regions in MA population (n=35) might be utilized to estimate the selection strength and the fitness effects on these coding genes.

Response: We thank you for your positive comments on our manuscript and putting forward many valuable suggestions. We have revised our manuscript according to the comments and suggestions by you and another reviewer in the revised manuscript.

We agreed with that more comparisons between MA populations and MA lines could help us better understand the process of selection, fitness, population size and mutation. Therefore, we have performed more comparisons between Heat MA lines and MA populations, including the frequencies of transition and transversion, coding regions, nonsynonymous mutations, and the estimation of genomic mutation rate affecting fitness. Moreover, based on the occurrence of mutations in coding regions, we further conducted the estimation of selection pressure on coding genes between MA lines and

MA populations using KaKs calculator. The Heat lines exhibited a Ka/Ks ratio of 0.92, whereas Heat population showed a Ka/Ks ratio > 1 (1.51). The new result supports our previous speculation that Heat population probably have subjected to stronger positive selection than MA lines.

In addition, we also discuss the different distribution patterns of mutations among all individuals between MA populations and MA lines. In each MA population, some mutations are shared by different individuals and retained in the final progeny; whereas the mutations accumulated in MA lines were diversely scattered in the individuals (no common mutations). This striking result was probably due to the selective sweep (or selection effects) in MA populations, resulting in the relative concentration of mutations in some genomic loci.

All these comparative results and analyses have been included in the corresponding Abstract, Results and Discussion sections in the revised manuscript.

Question 2: Under heat, the authors found mutations at DNA repair genes and also high mutation rates. These two observations could be connected and have nothing to do with heat. I wonder if the authors have considered a different scenario. Mutation occurred at DNA repair DNA just by chance in the heat MA line, and has nothing to do with the heat treatment. Because DNA repair gene was mutated, the mutation rate also increased. So the observed the mutation rate increases in heat was due to the mutation in DNA repair gene rather than heat. Please comment on this.

Response: Thank you for your comments. Indeed, we found only one heat MA line (not all heat lines) that exhibited mutations on a DNA repair gene. Therefore, it is not likely that the DNA repair gene is a mutation hotspot driven by heat treatment.

Previous “DNA repair” hypothesis suggests that the variation in fidelity of DNA repair may account for the variation in mutation rate (Baer et al. 2007). Therefore, we agree with your comments that the mutated DNA repair gene could be a possibility resulting in an increase in the mutation rate. On the other hand, in our results, we found that many Heat MA individuals have significantly increased mutation rates without mutations in

DNA repair genes. We speculate some possible reasons as follows: 1) Under heat stress, the mutation propensity of plants may not be directly driven by high temperature, but because some important biological processes had been significantly affected, including metabolism rate, cell division, DNA replication, and increased ROS (reactive oxygen species) level. In addition, the most basic chemical reactions of cellular activities, such as redox reaction, electron transfer, and macromolecule proteins or enzyme structure, would also be affected by high temperature. Therefore, in this regard, heat stress may act like a disruptor, interfering with the normal physiological, biochemical and growth process of plants, resulting in the increased disorders of cellular metabolism and instability in homeostasis. 2) When *Arabidopsis* plants grown under high temperature, heat stress induced more transcriptional activities of stress responsive genes, resulting in the increased probability of error in DNA unwinding and transcription. Moreover, heat stress would probably cause the heat stress-related impairment of DNA repair activity, for example, potential up-regulation of the error-prone polymerases typical of the SOS and SIM mechanisms identified in bacteria, yeast, and human cancer cells (Baer et al. 2007; Bindra, et al., 2011; Al Mamun et al. 2012; Shor et al. 2013). All these negative effects on plants are produced by heat stress, which may contribute to the increased mutation rates in Heat MA lines and populations.

References:

- Baer CF, Miyamoto MM, Denver DR. 2007. Mutation rate variation in multicellular eukaryotes: causes and consequences. *Nat Rev Genet* 8: 619–631.
- Bindra RS, Crosby ME, Glazer PM. 2007. Regulation of DNA repair in hypoxic cancer cells. *Cancer Metastasis Rev* 26: 249–260.
- Al Mamun AA, Lombardo M-J, Shee C, Lisewski AM, Gonzalez C, Lin D, Nehring RB, Saint-Ruf C, Gibson JL, Frisch RL, et al. 2012. Identity and function of a large gene network underlying mutagenic repair of DNA breaks. *Science* 338: 1344–1348.
- Shor E, Fox CA, Broach JR. 2013. The yeast environmental stress response regulates mutagenesis induced by proteotoxic stress. *PLoS Genet* 9: e1003680.

Question 3: Line 78. The authors define large duplications/deletions as del, but this is not consistent with line 161 where del refers to 1-3 short deletion. Please make changes and be consistent.

Response: In the Background, we have deleted the first “del” (Page 5, line 80) and retained “1-3bp short deletion” as “del” in the Results section (Page 9, line 163).

Question 4: Line 181-205 and figure 3. I would suggest including the standard error of the mean (SEM) in writing. SEM was shown in figure 3 but not written in line 181-205.

Response: As suggested, we have added the standard error of the mean (SEM) in the revised manuscript. Please see pages 10-11, lines 186-208.

Question 5: Line 352, 377 and lines 553-555. The authors used the word "ability" to describe the bias in mutation toward low gene density region and certain trinucleotide context. I found it a bit awkward to describe plants can control their mutation in a certain way. Changes in mutational parameters should be considered as an outcome the interplays between multiple evolutionary processes rather than controlled by the plant itself. I would suggest to modify the wording.

Response: We fully agree with your comments that accumulated mutational changes are the outcome of the interplays between multiple evolutionary processes rather than controlled by the plant itself. As per your suggestion, we have reworded the corresponding sentences.

1) We have changed the sentence in the original line 352 “This suggests that...ambient temperatures” to “This result suggests that multigenerational exposure of *A. thaliana* to high temperatures accelerates the accumulation of DNA mutations toward low gene density region compared to plants under ambient (Control) temperature”.

2) We have replaced the sentence in the original line 377 “However, the trinucleotides CCG (or GGC) and GCG (or CGC) appeared to have strong mutation abilities in all MA groups, regardless of temperature treatment” to “However, the trinucleotides CCG (or GGC) and GCG (or CGC) appeared to have high mutation rates in all MA groups, regardless of temperature treatment”.

3) We have revised the sentence in the original lines 553-535 “These results suggest that plants alter their genes to re-adjust their own physiological and developmental processes in response to long-term high temperature stress.” to “These results suggest that multigenerational exposure of *A. thaliana* to high temperatures promotes the mutations in stress response genes, thereby exerting influences on physiological and developmental processes in response to long-term high temperature stress.

Question 6: Figure S2 has a mismatched figure legend.

Response: We are sorry that the legend of Figure S2 was not presented correctly. In the revised manuscript, we have the necessary corrections. Please see the Additional file 2.

Question 7: Figure S3. It's not clear why the non-linear smooth function was used in the figure.

Response: In the original analysis, we had used both LOESS (locally weighted smoothing) and linear regression models (See Figure 1 below) to plot smooth function for the correlation between GC contents and mutation rates. The results showed that both models generally presented a flat fitted line, indicating that GC content had no significant impact on the mutation rate. Therefore, we showed only the correlation result for GC contents and observed mutation rates using the LOESS method (using loess in the stat_smooth function as part of the ggplot2 package). We have added the corresponding figure legend in the Additional file 2: Figure S3.

Figure 1. Relative mutation rates by local GC content across the *A. thaliana* genome of MA lines (Control D, Heat E, and Warming F) and populations (Control A, Heat B, and Warming C). A 1-kb bin size and 0.005 intervals of GC content were used. The figure was plotted using linear regression model.

Question 8: Additional file 3. Page 6. The citation for GSE118298 was incorrect.

Response: We are sorry for the mistake. In the revised manuscript, we have corrected the reference citation (Wang et al., 2020) and added the reference in the revised Additional file 3.

Second round of review

Reviewer 2

The revision has addressed all the comments and concerns. I recommend it for publication. A minor edit however is needed for abstract in lines 50 and 59. Please change population to MA population and change lines to MA lines. Please also spell out Mutation Accumulation for the first appearance.

Authors Response

Point-by-point responses to the reviewers' comments:

Reviewer #2: The revision has addressed all the comments and concerns. I recommend it for publication. A minor edit however is needed for abstract in lines 50 and 59. Please change population to MA population and change lines to MA lines. Please also spell out Mutation Accumulation for the first appearance.

Response: Thank you for your suggestion. We have changed “population” to “MA population” and “lines” to “MA lines” in the Abstract section. In addition, we added the “MA (mutation accumulation)” for the first appearance.